# Navigating Dataset Documentations in AI: A Large-Scale Analysis of Dataset Cards on Hugging Face

**Xinyu Yang** *
Cornell University
xy468@cornell.edu

**Weixin Liang***
Stanford University
wxliang@stanford.edu

**James Zou**
Stanford University
jamesz@stanford.edu

## Abstract

Advances in machine learning are closely tied to the creation of datasets. While data documentation is widely recognized as essential to the reliability, reproducibility, and transparency of ML, we lack a systematic empirical understanding of current dataset documentation practices. To shed light on this question, here we take Hugging Face – one of the largest platforms for sharing and collaborating on ML models and datasets – as a prominent case study. By analyzing all 7,433 dataset documentation on Hugging Face, our investigation provides an overview of the Hugging Face dataset ecosystem and insights into dataset documentation practices, yielding 5 main findings: (1) The dataset card completion rate shows marked heterogeneity correlated with dataset popularity: While 86.0% of the top 100 downloaded dataset cards fill out all sections suggested by Hugging Face community, only 7.9% of dataset cards with no downloads complete all these sections. (2) A granular examination of each section within the dataset card reveals that the practitioners seem to prioritize *Dataset Description* and *Dataset Structure* sections, accounting for 36.2% and 33.6% of the total card length, respectively, for the most downloaded datasets. In contrast, the *Considerations for Using the Data* section receives the lowest proportion of content, accounting for just 2.1% of the text. (3) By analyzing the subsections within each section and utilizing topic modeling to identify key topics, we uncover what is discussed in each section, and underscore significant themes encompassing both technical and social impacts, as well as limitations within the *Considerations for Using the Data* section. (4) Our findings also highlight the need for improved accessibility and reproducibility of datasets in the *Usage* sections. (5) In addition, our human annotation evaluation emphasizes the pivotal role of comprehensive dataset content in shaping individuals' perceptions of a dataset card's overall quality. Overall, our study offers a unique perspective on analyzing dataset documentation through large-scale data science analysis and underlines the need for more thorough dataset documentation in machine learning research.

## 1 Introduction

Datasets form the backbone of machine learning research (Koch et al., 2021). The proliferation of machine learning research has spurred rapid advancements in machine learning dataset development, validation, and real-world deployment across academia and industry. Such growing availability of ML datasets underscores the crucial role of proper documentation in ensuring transparency, reproducibility, and data quality in research (Haibe-Kains et al., 2020; Stodden et al., 2018; Hutson, 2018). Documentation provides details about the dataset, including sources of data, methods used to collect it, and preprocessing or cleaning that was performed. This information holds significant value for dataset users, as it facilitates a quick understanding of the dataset's motivation and its overall scope. These insights are also crucial for fostering responsible data sharing and promoting interdisciplinary collaborations.

---

*These authors contributed equally to this work.

Despite numerous studies exploring the structure and content of dataset cards across various research domains (Afzal et al., 2020; Gebru et al., 2021; Papakyriakopoulos et al., 2023; Barman et al., 2023; Costa-jussà et al., 2020), there remains a notable gap in empirical analyses of community norms and practices for dataset documentation. This knowledge gap is significant because adherence to community norms and the quality of dataset documentation directly impact the transparency, reliability, and reproducibility in the field of data-driven research. For instance, inadequate dataset descriptions, structural details, or limitations can hinder users from utilizing the dataset appropriately, potentially resulting in misuse or unintended consequences; the absence of information on data cleaning and readiness assessment practices in data documentation limits dataset reusability and productivity gains. Furthermore, without a systematic analysis of current dataset documentation practices, we risk perpetuating insufficient documentation standards, which can impede efforts to ensure fairness, accountability, and equitable use of AI technologies.

To address this question, we conducted a comprehensive empirical analysis of dataset cards hosted on Hugging Face, one of the largest platforms for sharing and collaborating on ML models and datasets, as a prominent case study. Dataset cards on the Hugging Face platform are Markdown files that serve as the README for a dataset repository. While several open-source platforms also facilitate the sharing of ML datasets, such as Kaggle, Papers with Code, and GitHub, we chose Hugging Face for two primary reasons. Firstly, it stands out as one of the most popular platforms for developers to publish, share, and reuse ML-based projects, offering a vast repository of ML datasets for study. Secondly, Hugging Face is one of the few open-source platforms that offer an official dataset card template. This feature not only enhances the accessibility and user-friendliness of the dataset card community but also makes the analysis process more efficient and informative.

By analyzing all 7,433 dataset documentation hosted on Hugging Face, our investigation provides an overview of the Hugging Face dataset ecosystem and insights into dataset documentation practices. Based on our research findings, we emphasize the importance of comprehensive dataset documentation and offer suggestions to practitioners on how to write documentation that promotes reproducibility, transparency, and accessibility of their datasets, which can help to improve the overall quality and usability of the dataset community. Our study aims to bridge the notable gap in the community concerning data documentation norms, taking the first step toward identifying deficiencies in current practices and offering guidelines for enhancing dataset documentation.

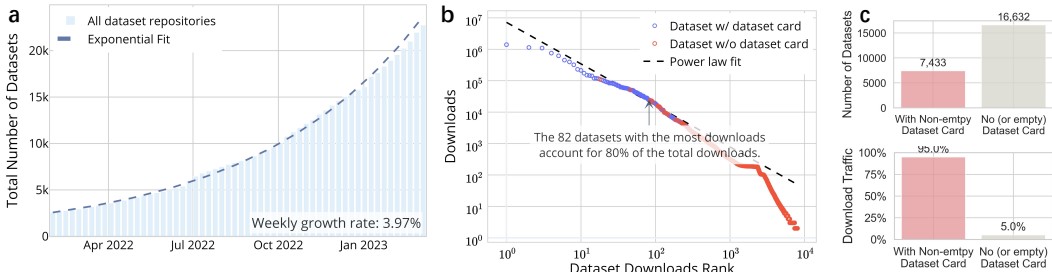

Figure 1: **Systematic Analysis of 24,065 Datasets Hosted on Hugging Face.** (**a**) *Exponential Growth of Datasets:* The Hugging Face platform has seen a remarkable surge in the number of datasets, with the count doubling approximately every 18 weeks. (**b**) *Power Law in Dataset Usage*: Dataset downloads on Hugging Face follow a power-law distribution, as indicated by the linear relationship on the log-log plot. The top 82 datasets account for 80% of the total downloads; datasets with documentation dominate the top downloaded datasets. (**c**) *Documentation Associated with Usage:* Despite only 30.9% of dataset repositories (7,433 out of 24,065) featuring non-empty dataset cards, these datasets account for an overwhelming 95.0% of total download traffic on the platform.

## 2 OVERVIEW

> **Finding**
>
> - **Exponential Growth of Datasets:** The number of datasets on Hugging Face doubles every 18 weeks.
> - **Documentation Associated with Usage:** 95.0% of download traffic comes from the 30.9% of datasets with documentation.

**Exponential Growth of Datasets** Our analysis encompasses 24,065 dataset repositories on Hugging Face uploaded by 7,811 distinct user accounts as of March 16th, 2023 (see **Table. S5** for varying documentation practices by creators). The number of datasets exhibits exponential growth, with a weekly growth rate of 3.97% and a doubling time of 18 weeks (**Fig. 1**$a$). As a sanity check, the number of dataset repositories reached 35,973 by May 23rd, 2023, confirming the exponential trend.

**Power Law in Dataset Usage** Although Hugging Face has seen a significant increase in the number of dataset repositories, our analysis reveals a significant imbalance in dataset downloads, which follows a power law distribution. This means that a small proportion of the most popular datasets receive the majority of the downloads, while the vast majority of datasets receive very few downloads. In fact, our analysis shows that just the 82 datasets with the most downloads account for 80% of total downloads (**Fig. 1**$b$). **Fig. S4** further demonstrates that the power law distribution persists across various task domains, even with the varied number of datasets within each domain.

**Documentation Associated with Usage** Despite the importance of dataset cards, only 58.2% (14,011 out of 24,065 dataset repositories contributed by 4,782 distinct user accounts) include dataset cards as Markdown README.md files within their dataset repositories. Among these, 6,578 dataset cards are empty, resulting in only 30.9% (7,433 out of 24,065 dataset repositories contributed by 1,982 distinct user accounts) featuring non-empty dataset cards (**Fig. 1**$c$). As illustrated in **Fig. 1**$b$, dataset cards are prevalent among the most downloaded datasets. Notably, datasets with non-empty dataset cards account for 95.0% of total download traffic, underscoring a potential positive correlation between dataset cards and dataset popularity. For the rest of the paper, we focus our analyses on these 7,433 non-empty dataset cards. We sort these non-empty dataset cards based on the number of downloads for the corresponding datasets. So top $k$ dataset cards (e.g. $k = 100$) refer to the dataset cards corresponding to the $k$ most downloaded datasets.

## 3 STRUCTURE OF DATASET DOCUMENTATIONS

> **Finding**
>
> - **The dataset card completion rate shows marked heterogeneity correlated with dataset popularity:** While 86.0% of the top 100 downloaded datasets fill out all sections suggested by the Hugging Face community, only 7.9% of dataset cards with no downloads complete all these sections.

| Section Title | Subsection Title | Description |
|---|---|---|
| Dataset Description | Dataset Summary | A brief summary of the dataset, including its intended use, supported tasks, an overview of how and why the dataset was created, etc. |
| | Supported Tasks and Leaderboards | Brief description of the tag, metrics, and suggested models of the dataset. |
| | Languages | The languages represented in the dataset. |
| Dataset Structure | Data Instances | JSON-formed example and description of a typical instance in the dataset. |
| | Data Fields | List and describe the fields present in the dataset. Mention their data type, and whether they are used as input or output in any of the tasks the dataset currently supports. |
| | Data Splits | Criteria for splitting the data, descriptive statistics for the features, such as size, average length, etc. |
| Dataset Creation | Curation Rationale | Motivation for the creation of the dataset. |
| | Source Data | The source data (e.g. news text and headlines, social media posts, translated sentences, etc.), including the data collection process, and data producer. |
| | Annotations | Annotation process, annotation tools, annotators, etc. |
| | Personal and Sensitive Information | Statement of whether the dataset contains other data that might be considered sensitive (e.g., data that reveals racial or ethnic origins, financial or health data, etc.). |
| Considerations for Using the Data | Social Impact of Dataset | Discussion of the ways the use of the dataset will impact society. |
| | Discussion of Biases | Descriptions of specific biases that are likely to be reflected in the data |
| | Other Known Limitations | Other limitations of the dataset, like annotation artifacts. |
| Additional Information | Dataset Curators | The people involved in collecting the dataset and their affiliation(s) |
| | Licensing Information | The license and link to the license webpage if available. |
| | Citation Information | The BibTex-formatted reference for the dataset. |
| | Contributions | 'Thanks to @github-username for adding this dataset.' |

Table 1: **Community-Endorsed Dataset Card Structure.** This table shows the sections and their suggested subsections provided by the Hugging Face community, along with their descriptions. For more information, please refer to `https://github.com/huggingface/datasets/blob/main/templates/README_guide.md`.

**Community-Endorsed Dataset Card Structure** Grounded in academic literature (Mitchell et al., 2019) and official guidelines from Hugging Face (HuggingFace, 2021), the Hugging Face community provides suggestions for what to write in each section. This community-endorsed dataset card provides a standardized structure for conveying key information about datasets. It generally contains 5 sections: *Dataset Description*, *Dataset Structure*, *Dataset Creation*, *Considerations for Using the Data*, and *Additional Information* (**Table. 1**). To examine the structure of dataset cards, we used a pipeline that detects exact word matches for each section title. We then identified the section titles and checked whether they had contents (**Appendix B.1**). If a dataset card had all five sections completed, we considered it to be following the community-endorsed dataset card.

**Adherence to Community-Endorsed Guidelines Correlates with Popularity** Our evaluation found that popular datasets have better adherence to the dataset card community-endorsed dataset card structure. As illustrated in **Fig. 2**, compliance with the template varies significantly among datasets with different download counts. Among the 7,433 dataset cards analyzed, 86.0% of the top 100 downloaded dataset cards have completed all five sections of the community-endorsed dataset card, while

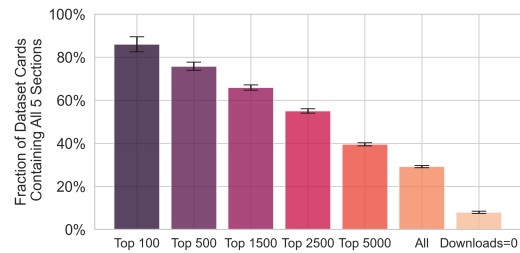

Figure 2: **Highly downloaded datasets consistently show better compliance with the community-endorsed documentation structure.**

only 7.9% of dataset cards with no downloads follow it. **Fig. S5** further reveals that popular dataset cards achieve higher completion in all Hugging Face-recommended sections. This implies a potential correlation between adherence to community-endorsed guidelines and dataset popularity.

## 4 PRACTITIONERS EMPHASIZE DESCRIPTION AND STRUCTURE OVER SOCIAL IMPACT AND LIMITATIONS

**Finding**

- **Practitioners seem to prioritize on *Dataset Description* and *Dataset Structure* sections**, which account for 36.2% and 33.6% of the total card length, respectively, on the top 100 most downloaded datasets.
- **In contrast, the *Considerations for Using the Data* section receives the lowest proportion of content, just 2.1%.** The *Considerations for Using the Data* section covers the social impact of datasets, discussions of biases, and limitations of datasets.

**Social Impact, Dataset Limitations and Biases are Lacking in Most Documentations** Following the community-endorsed dataset card, we conducted an analysis to determine the level of emphasis placed on each section. **Fig. 3***b* shows the word count distribution among the top 100 downloaded dataset cards, revealing their high level of comprehensiveness: 91.0% of them have a word count exceeding 200. We step further into these dataset cards to examine the emphasis placed on each section. We calculated the word count of each section and its proportion to the entire dataset card. As shown in **Fig. 3***c*, the *Dataset Description* and *Dataset Structure* sections received the most attention, accounting for 36.2% and 33.6% of the dataset card length, respectively. On the other hand, the *Considerations for Using the Data* section received a notably low proportion of only 2.1%.

**Section Length Reflects Practitioner Attention** The length of sections within dataset cards is reflective of practitioner attention, and it varies significantly based on the popularity of the dataset. Highly downloaded datasets tend to have more comprehensive and longer dataset cards (**Fig. 3***a*), with an emphasis on the *Dataset Description* and *Dataset Structure* sections (**Fig. 3***d*). Conversely, less popular datasets have shorter cards (**Fig. 3***a*) with a greater emphasis on the *Additional Information* section (**Fig. 3***d*). Despite this, sections such as *Dataset Creation* and *Considerations for Using the Data* consistently receive lower attention, regardless of download rates (**Fig. 3***d*). This suggests a need to promote more comprehensive documentation, particularly in critical sections, to enhance dataset usage and facilitate ethical considerations.

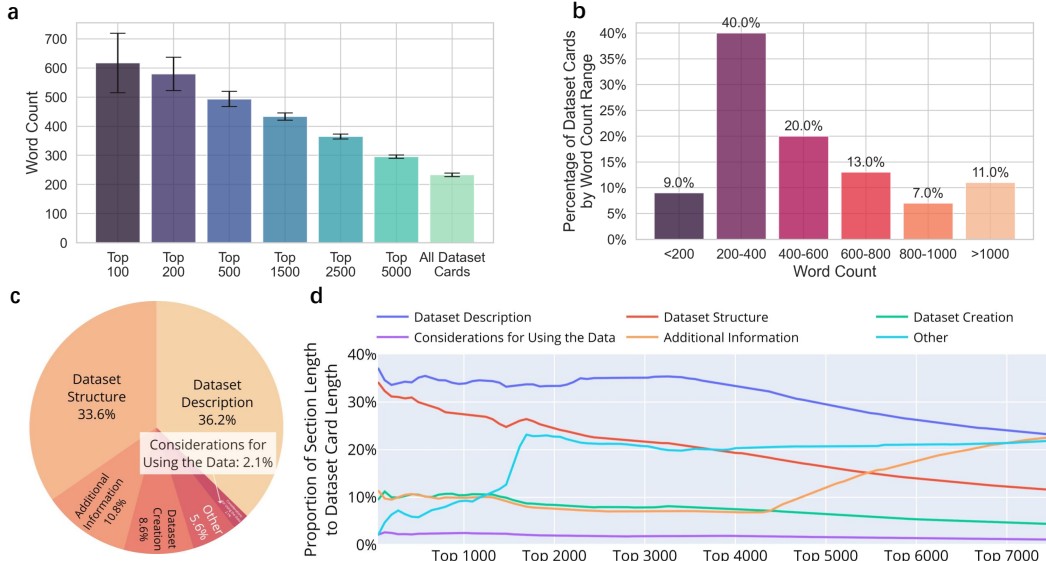

Figure 3: **Section Length Reflects Practitioner Attention.** (**a**) *Popularity Correlates with Documentation Length:* The top downloaded dataset cards are longer, indicating that they contain more comprehensive information. (**b**) Distribution of Word Count Among Top 100 Downloaded Dataset Cards (**c**) *Section Length Proportions in Top 100 Downloaded Dataset Cards:* The *Dataset Description* and *Dataset Structure* sections dominate in the top 100 downloaded dataset cards, with proportions of 36.2% and 33.6%, respectively. In contrast, the *Considerations for Using the Data* section receives the least attention, with a proportion of only 2.1%. (**d**) *Section Length Proportion Changes over Downloads:* The section length proportion changes over downloads, with *Dataset Description* and *Dataset Structure* decreasing in length, and *Additional Information* and *Other* increasing. Notably, there is a consistently low emphasis placed on the *Dataset Creation* and *Considerations for Using the Data* sections across all dataset cards with different downloads.

## 5 UNDERSTANDING CONTENT DYNAMICS IN DATASET DOCUMENTATION

> **Finding**
>
> - **Strong Community Adherence to Subsection Guidelines:** Practitioners contributing to the Hugging Face community exhibit high compliance with standards, filling out 14 of the 17 recommended subsections across five main sections at a rate exceeding 50%.
> - **Emergence of the *Usage* Section Beyond the Community Template:** Surprisingly, 33.2% of dataset cards includes a *Usage* section. The community template does not include such *Usage* section in its current form and should include one in the future.

**Section Content Detection Pipeline**   To gain a deeper understanding of the topics discussed in each section, we conducted a content analysis within each section of the community-endorsed dataset card structure, which includes suggested *subsections* within the five main sections. We used exact keyword matching to identify the corresponding subsections and calculate their filled-out rates. **Fig. 4** shows that 14 out of 17 subsections have filled-out rates above 50%, indicating adherence to the community-endorsed dataset cards.

**Limitation Section is Rare, but Long if it Exists**   The *Considerations for Using the Data* section (i.e., limitation section), despite being frequently overlooked and often left empty by practitioners, holds particular significance. When this section is included, it tends to adhere well to community guidelines, with subsections having a completion rate exceeding 50% and a reasonably substantial word count (98.2 words). This suggests that this section has the potential to provide valuable insights and guidance. This motivates our use of topic modeling to identify key discussion topics within this section, potentially aiding practitioners in crafting meaningful content.

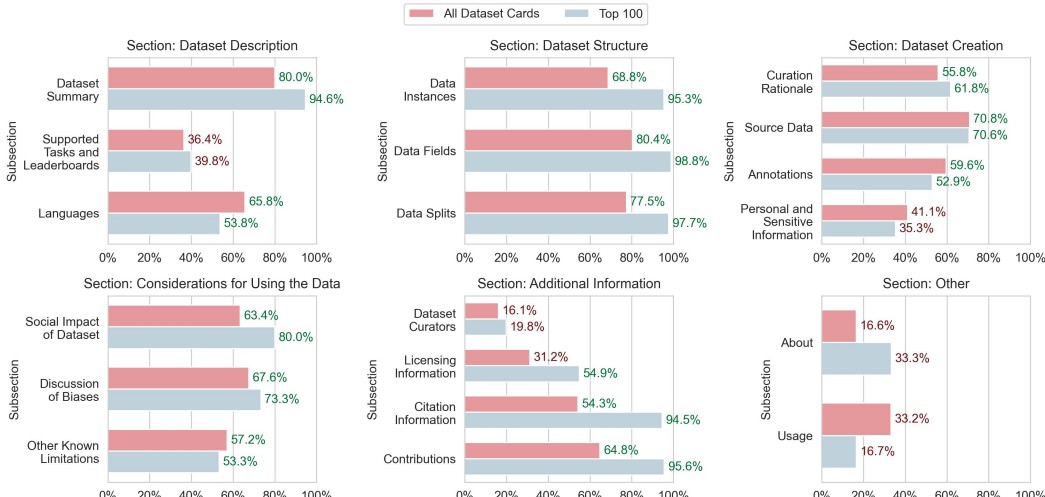

Figure 4: **Highlighting the Hugging Face Community's Compliance with Subsection Guidelines.** This figure shows subsection filled-out rates within different sections, stratified by download counts. Each section has multiple subsections, with bars representing the filled-out rate of each subsection. Green texts indicate filled-out rates above 50%, while red texts indicate rates below 50%. Of the 17 subsections within the five sections of the community-endorsed dataset, 14 have filled-out rates above 50%.

| a | Social Impact of Dataset |
|---|---|
| **Topic** | **Representative Sentences** |
| Technical or Research Scope | • Adding a Spanish resource may help others to improve their research and educational activities.
• The creation of the dataset contributes to expanding the scope of NLP research to under-explored languages across the world. |
| Social Scope or Background | • This dataset can be used to gain insights into the social, cultural, and political views of people in African countries.
• If this matter isn't tackled with enough urgency, we might see the rise of a new dark era in Latin America politics, where many unscrupulous parties and people will manage to gain power and control the lives of many people. |

| b | Discussion of Biases |
|---|---|
| **Topic** | **Representative Sentences** |
| Subpopulation Biases | • Gender speakers distribution is imbalanced, percentage of female speakers is mostly lower than 50% across languages.
• The social biases of the time in terms of race, sex, gender, etc. might be encountered in this dataset. |
| Biases from Collection Procedure | • With respect to the potential risks, we note that the subjectivity of human annotation would impact on the quality of the dataset.
• In terms of data collection, by using keywords and user mentions, we are introducing some bias to the data, restricting our scope to the list of keywords and users we created. |

| c | Other Known Limitations |
|---|---|
| **Topic** | **Representative Sentences** |
| Data Quality | • The nature of the task introduce a variability in the quality of the target translations.
• A number of errors, omissions and inconsistencies are expected to be found within the corpus. |
| Processing Limitation | • Our augmentation process can sometimes create nonexistent versions of real people.
• Satellite annotation is not as accurate for pixel-level representation due to single-point annotations. |

Figure 5: **Key Topics in *Considerations for Using the Data* through Topic Modeling Analysis.** This figure displays the outcomes of the topic modeling assessment on the contents of the (**a**) *Social Impact of Dataset* Subsection, (**b**) *Discussion of Biases* Subsection, and (**c**) *Other Known Limitations* Subsection. Each panel illustrates the human-assigned topic label and representative sentences for each section. Topics are generated by Latent Dirichlet Allocation (LDA).

**Limitation Section Covers Diverse and Crucial Topics**   The *Considerations for Using the Data* section (i.e., limitation section) encompasses diverse and crucial topics. The Hugging Face community emphasizes three major themes within this section: *Social Impact of Dataset*, *Discussion of Biases*, and *Other Known Limitations*.

The *Social Impact of Dataset* aspect explores not only societal implications but also the potential benefits to technology and research communities. In this section, practitioners discuss issues like

how the dataset can expand the scope of NLP research (Armstrong et al., 2022), and increase access to natural language technology across diverse regions and cultures (Tache et al., 2021). Additionally, the subsection covers sensitive topics related to politics, ethics, and culture within the social scope.

*Discussion of Biases* delves into subpopulation bias and data collection biases, highlighting the importance of addressing bias-related issues. Previous research have identified numerous technical and social biases such as subgroup bias (Buolamwini & Gebru, 2018), data collection bias (Wang et al., 2019), and label bias (Jiang & Nachum, 2020). Our topic modeling results reveal that two primary biases are discussed by practitioners in this subsection. The first is subpopulation bias, which includes biases related to gender, age, or race. For instance, an audio dataset (Nsoesie & Galea, 2022) notes that female speakers are underrepresented, comprising less than 50% of the dataset. The second major bias arises from the data collection process, specifically the annotation process, which is often a significant bottleneck and source of errors.

Lastly, *Other Known Limitations* focuses on technical limitations, particularly data quality and processing limitations. This comprehensive coverage underscores the multifaceted nature of considerations related to dataset usage. Data quality is often a focus in other disciplines, such as the social sciences and biomedicine, and there are many insights to draw upon (Paullada et al., 2021; Fedorov, 2010; Fan & Geerts, 2012). Meanwhile, processing limitations encompass a broader range of issues beyond biases from the collection procedure, such as inaccuracies or the absence of some data points.

**Emergence of the *Usage* Section Beyond the Community Template**  While Hugging Face's community-endorsed dataset card structure comprises five main sections, there are instances where practitioners encounter valuable information that doesn't neatly fit into these sections. These additional sections, referred to as *Other* sections, can contain important content. Notably, among these *Other* sections, discussions related to *Usage* emerge as a frequent (nearly one-third of the time, 33.2%) and significant theme. These *Usage* sections offer a diverse range of information, including details on downloading, version specifications, and general guidelines to maximize the dataset's utility. This highlights the importance of considering content that falls outside the predefined template and suggests a potential area for improvement in dataset card templates.

**Quantifying the Impact of *Usage* Section on Dataset Downloads**  To assess the influence of a *Usage* section in dataset documentation, we conducted a counterfactual analysis experiment (**Appendix. C**). We trained a BERT (Devlin et al., 2018) model using dataset card content and download counts, which were normalized to fall within the range of [0, 1] for meaningful comparisons. When a dataset card that initially included a *Usage* section had this section removed, there was a substantial decrease of 1.85% in downloads, with statistical significance. This result underscores the significant impact of the *Usage* section in bolstering dataset accessibility and popularity, emphasizing its pivotal role in enhancing the documentation and usability of datasets.

## 6 ANALYZING HUMAN PERCEIVED DATASET DOCUMENTATION QUALITY

> **Finding**
>
> - **Our human annotation evaluation emphasizes the pivotal role of *comprehensive dataset content* in shaping individuals' perceptions of a dataset card's overall quality.**

**Human Annotations for Comprehensive Evaluation of Dataset Card Quality**  We utilized human annotations to evaluate the quality of dataset cards, considering seven distinct aspects, drawing from prior research in dataset documentation literature and the Hugging Face community-endorsed dataset card  (Afzal et al., 2020; Gebru et al., 2021; Papakyriakopoulos et al., 2023; Barman et al., 2023; Costa-jussà et al., 2020): (1) Structural Organization, (2) Content Comprehensiveness, (3) Dataset Description, (4) Dataset Structure, (5) Dataset Preprocessing, (6) Usage Guidance, and (7) Additional Information. While Dataset Description, Dataset Structure, and Additional Information can be found in sections of community-endorsed dataset cards, we added evaluation aspects highlighted in the literature, like aspects that constitute the overall presentation (Structural Organization and Content Comprehensiveness), Data Preprocessing and Usage Guidance. To conduct this assessment, we randomly selected a subset containing 150 dataset cards and engaged five human annotators. These

annotators were tasked with evaluating each dataset card across these seven aspects and providing an overall quality score within a range of 5 (**Appendix B.2**). The overall quality is assessed through the subjective perception of human annotators, taking into account the seven aspects as well as their overall impression. This evaluation approach aims to provide a comprehensive assessment of dataset card quality, reflecting the importance of these aspects in effective dataset documentation.

**Human Perception of Documentation Quality Strongly Aligns with Quantitative Analysis**
Human annotation evaluation of dataset cards shows varying scores across different aspects. While Dataset Description (2.92/5), Structural Organization (2.82/5), Data Structure (2.7/5), and Content Comprehensiveness (2.48/5) received relatively higher scores, areas like Data Preprocessing (1.21/5) and Usage Guidance (1.14/5) scored lower. This aligns with the quantitative analysis that indicates a greater emphasis on the *Dataset Description* and *Dataset Structure* sections. Notably, even the highest-scoring aspect, Dataset Description, falls below 60% of the highest possible score, indicating room for improvement in dataset documentation.

**Content Comprehensiveness has the strongest positive correlation with the overall quality of a dataset card (Coefficient: 0.3935, p-value: 3.67E-07)**, emphasizing the pivotal role of comprehensive dataset content in shaping individuals' perceptions of a dataset card's overall quality. Additionally, aspects like Dataset Description (Coefficient: 0.2137, p-value: 3.04E-07), Structural Organization (Coefficient: 0.1111, p-value: 2.17E-03), Data Structure (Coefficient: 0.0880, p-value: 6.49E-03), and Data Preprocessing (Coefficient: 0.0855, p-value: 2.27E-03) also significantly contribute to people's evaluations of dataset documentation quality. Moreover, the length of a dataset card is positively related to Content Comprehensiveness (p-value: 1.89E-011), reinforcing the importance of detailed documentation in enhancing dataset quality and usability.

## 7 RELATED WORKS

Dataset has long been seen as a significant constraint in the realm of machine learning research (Halevy et al., 2009; Sun et al., 2017). The process of creating datasets remains arduous and time-intensive, primarily due to the costs of curation and annotation (IBM, 2020). Moreover, the quality of data assumes a pivotal role in shaping the outcomes of machine learning research (Liang et al., 2022). Consequently, a profound understanding of datasets is indispensable in the context of machine learning research, and this understanding is most effectively conveyed through comprehensive dataset documentation.

A long-standing problem in the literature is that there is no industry standard being formed about data documentation. Therefore, much existing work in the literature has been in exploring, conceptualizing and proposing different dataset documentation frameworks. Data-focused tools such as datasheets for datasets and data nutrition labels have been proposed to promote communication between dataset creators and users, and address the lack of industry-wide standards for documenting AI datasets (Bender & Friedman, 2018; Bender et al., 2021; Pushkarna et al., 2022; Gebru et al., 2021; Holland et al., 2018; Chmielinski et al., 2022; Papakyriakopoulos et al., 2023). Additionally, there are studies that concentrate on leveraging human-centered methods to scrutinize the design and evaluation aspects of dataset documentation (Fabris et al., 2022; Mahajan & Shaikh, 2021; Hanley et al., 2020; Hutiri et al., 2022). In the library domain, numerous works have proposed methods to tackle the absence of universally accepted guidelines for publishing library-linked data. These efforts are aimed at enhancing data quality, promoting interoperability, and facilitating the discoverability of data resources (Villazón-Terrazas et al., 2011; Hidalgo-Delgado et al., 2017; Abida et al., 2020). These tools and frameworks provide detailed information on the composition, collection process, recommended uses, and other contextual factors of datasets, promoting greater transparency, accountability, and reproducibility of AI results while mitigating unwanted biases in AI datasets. Additionally, they enable dataset creators to be more intentional throughout the dataset creation process. Consequently, datasheets and other forms of data documentation are now commonly included with datasets, helping researchers and practitioners to select the most appropriate dataset for their particular needs.

Despite the proliferation of dataset documentation tools and the growing emphasis on them, the current landscape of dataset documentation remains largely unexplored. In this paper, we present

a comprehensive analysis of AI dataset documentation on Hugging Face to provide insights into current dataset documentation practices.

## 8  DISCUSSION

In this paper, we present a comprehensive large-scale analysis of 7,433 AI dataset documentation on Hugging Face. The analysis offers insights into the current state of adoption of dataset cards by the community, evaluates the effectiveness of current documentation efforts, and provides guidelines for writing effective dataset cards. Overall, our main findings cover 5 aspects:

- *Varied Adherence to Community-Endorsed Dataset Card:* We observe that high-downloaded dataset cards tend to adhere more closely to the community-endorsed dataset card structure.

- *Varied Emphasis on Sections:* Our analysis of individual sections within dataset cards reveals that practitioners place varying levels of emphasis on different sections. For instance, among the top 100 downloaded dataset cards, *Dataset Description* and *Dataset Structure* sections receive the most attention. In contrast, the *Considerations for Using the Data* section garners notably lower engagement across all downloads, with only approximately 2% of dataset cards containing this section. This discrepancy can be attributed to the section's content, which involves detailing limitations, biases, and the societal impact of datasets – a more complex and nuanced endeavor. An internal user study conducted by Hugging Face (HuggingFace, 2022) also identified the *Limitation* section within this category as the most challenging to compose.

- *Topics Discussed in Each Section:* Our examination of subsections within each section of dataset cards reveals a high completion rate for those suggested by the Hugging Face community. This highlights the effectiveness of the community-endorsed dataset card structure. In particular, our study places a special focus on the *Considerations for Using the Data* section, employing topic modeling to identify key themes, including technical and social aspects of dataset limitations and impact.

- *Importance of Including Usage Sections:* We observe that many dataset card creators go beyond the recommended structure by incorporating *Usage* sections, which provide instructions on effectively using the dataset. Our empirical experiment showcases the potential positive impact of these *Usage* sections in promoting datasets, underscoring their significance.

- *Human Evaluation of Dataset Card Quality:* Our human evaluation of dataset card quality aligns well with our quantitative analysis. It underscores the pivotal role of Content Comprehensiveness in shaping people's assessments of dataset card quality. This finding offers clear guidance to practitioners, emphasizing the importance of creating comprehensive dataset cards. Moreover, we establish a quantitative relationship between Content Comprehensiveness and the word length of dataset cards, providing a measurable method for evaluation.

**Limitations and Future Works**    Our analysis of ML dataset documentation relies on the distinctive community-curated resource, Hugging Face, which may introduce biases and limitations due to the platform's structure and coverage. For example, Hugging Face's NLP-oriented concentration could introduce biases into the dataset categories. However, our method is transferable and could easily be reproduced for another platform, facilitating future studies (**Appendix. E**). Additionally, our analysis of completeness and informativeness is based on word count and topic modeling, which may not fully capture the nuances of the documentation. Furthermore, measuring dataset popularity based on downloads alone may not fully reflect the dataset's impact. Future research could consider additional factors, such as the creation time of the dataset and research area of the dataset (**Appendix. D**). Lastly, our human evaluation serves as a preliminary evaluation. Future analyses could involve a more diverse group of annotators with varying backgrounds and perspectives.

**Research Significance**    To summarize, our study uncovers the current community norms and practices in dataset documentation, and demonstrates the importance of comprehensive dataset documentation in promoting transparency, accessibility, and reproducibility in the AI community. We hope to offer a foundation step in the large-scale empirical analysis of dataset documentation practices and contribute to the responsible and ethical use of AI while highlighting the importance of ongoing efforts to improve dataset documentation practices.

REPRODUCIBILITY STATEMENT

We have assembled a collection of dataset cards as a community resource, which includes extracted metadata such as the number of downloads and textual analyses. This resource along with our analysis code can be accessed at `https://github.com/YoungXinyu1802/HuggingFace-Dataset-Card-Analysis`. The Hugging Face datasets can be accessed through the Hugging Face Hub API, which is available at `https://huggingface.co/docs/huggingface_hub/package_reference/hf_api`.

ACKNOWLEDGMENTS

We thank Yian Yin and Nazneen Rajani for their helpful comments and discussions. J.Z. is supported by the National Science Foundation (CCF 1763191 and CAREER 1942926), the US National Institutes of Health (P30AG059307 and U01MH098953) and grants from the Silicon Valley Foundation and the Chan-Zuckerberg Initiative.

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

## A    ILLUSTRATIONS FOR DATASET CARDS SUGGESTED BY HUGGING FACE COMMUNITY

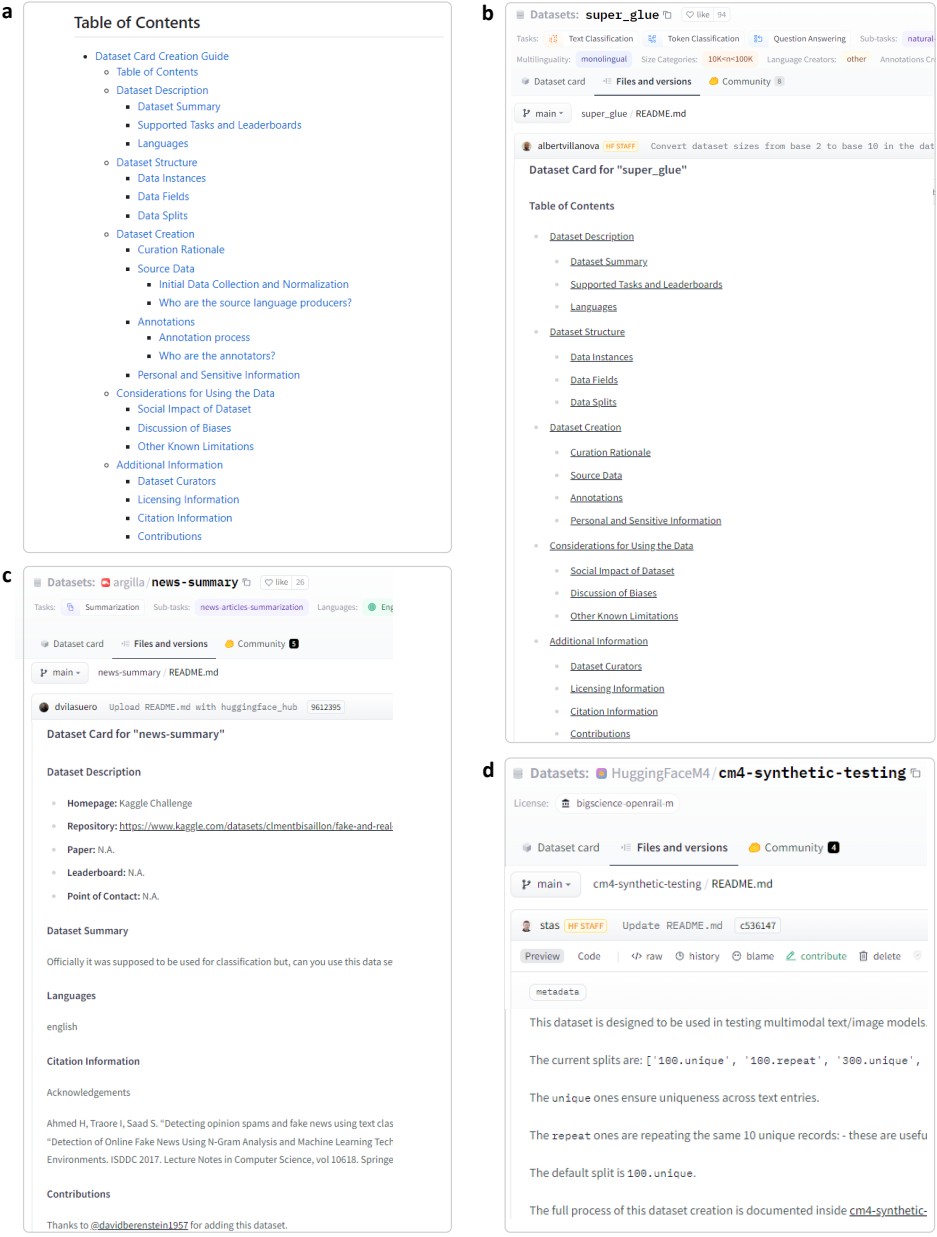

Figure S1: **Illustration of Adherence to Community-Endorsed Dataset Card.** (**a**) *Community-Endorsed Dataset Card Struture:* Hugging Face community provides a suggested dataset card structure, which contains five main sections: *Dataset Description*, *Dataset Structure*, *Dataset Creation*, *Considerations for Using the Data*, and *Additional Information*. (**b**) *Example of a Dataset Card Conforming to the Community Guidelines:* A dataset card is considered to conform to the community guidelines when it includes the five main sections outlined in the community guidelines, with the corresponding content provided for each section. (**c**) *Example of Dataset Cards Not Following Community Guidelines (1):* A dataset card is considered non-conforming if it omits any of the five main sections provided in the suggested dataset card structure. (**d**) *Example of Dataset Cards Not Following Community Guidelines (2):* This dataset card contains only a few words and does not follow the structure at all.

## B   METHOD

### B.1   ACCESSING AND PARSING DATASET CARDS

In this work, we analyze datasets hosted on Hugging Face, a popular platform that provides a wealth of tools and resources for AI developers. One of its key features is the Hugging Face Hub API, which grants access to a large library of pre-trained models and datasets for various tasks. With this API, we obtained all 24,065 datasets hosted on the Hub as of March 16th, 2023.

Dataset cards are Markdown files that serve as the README for a dataset repository. They provide information about the dataset and are displayed on the dataset's homepage. We downloaded all dataset repositories hosted on Hugging Face and extracted its README file to get the dataset cards. For further analysis of the documentation content, we utilized the Python package mistune (`https://mistune.readthedocs.io/en/latest/`) to parse the README file and extract the intended content. The structure of dataset cards typically consists of five sections: *Dataset Description*, *Dataset Structure*, *Dataset Creation*, *Additional Information*, and *Considerations for Using the Data*, as recommended by Hugging Face community. Examples of dataset cards, as shown in **Fig. S1**, illustrate the essential components and information provided by dataset cards. We identified and extracted different types of sections through parsing and word matching of the section heading. A significant 84% of the section titles in the 7,433 dataset cards matched one of the 27 titles suggested by the HuggingFace community using the exact keyword matching. This strong alignment underscores the effectiveness of exact keyword matching as an analytical tool.

### B.2   HUMAN-ANNOTATED DATASET CARD EVALUATION METHODOLOGY AND CRITERIA

We conducted an evaluation on a sample of 150 dataset cards from a total of 7,433. The assessment involved five human annotators to evaluate the dataset cards, who are PhD students with a solid background in AI fields such as NLP, Computer Vision, Human-AI, ML, and Data Science. Their extensive experience with datasets ensured a deep understanding of dataset documentation. To confirm the reliability of our evaluation, we randomly sampled 30 dataset cards for the annotators to assess and achieved an Intraclass Correlation Coefficient (ICC) of 0.76, which is considered a good agreement (Koo & Li, 2000). This high level of agreement, combined with the annotators' deep expertise in AI research, substantially reinforces the trustworthiness of the annotation results. We focused on seven key aspects of the dataset cards drawing from prior research in dataset documentation and the Hugging Face community-endorsed dataset card:

| Aspect | Description |
| --- | --- |
| Structural Organization | How well is the documentation structured with headings, sections, or subsections? |
| Content Comprehensiveness | How comprehensive is the information provided in the documentation? |
| Dataset Description | How effectively does the documentation describe the dataset? |
| Dataset Preprocessing | How well does the documentation describe any preprocessing steps applied to the data? |
| Usage Guidance | How well does the documentation offer guidance on using the dataset? |
| Additional Information | How well does the documentation provide extra details such as citations and references? |

Table S1: **Descriptions of Evaluation Aspects**

Each aspect received a score on a scale from 0 to 5, with the following score metrics:

| Score | Description | Comment |
|---|---|---|
| 5 | Exceptionally comprehensive and effective | Covers all subsections in detail |
| 4 | Very good and thorough | Includes many subsections comprehensively |
| 3 | Moderately satisfactory | Covers some subsections adequately |
| 2 | Insufficient | Provides a basic, general overview |
| 1 | Poor and inadequate | Offers minimal, vague content |
| 0 | Absent | Lacks relevant content |

Table S2: **Metrics of the Scores**

## C    ADDITIONAL ANALYSIS OF *Usage* SECTION

Among 7,433 dataset cards, there are 567 dataset cards uploaded by 52 distinct practitioners that contain a *Usage* section, instructing how to use the dataset through text and codes. A specific example of *Usage* section is from ai4bharat/naamapadam, which has 469 downloads and has a *Usage* section to instruct how to use the dataset (**Fig. S2**).

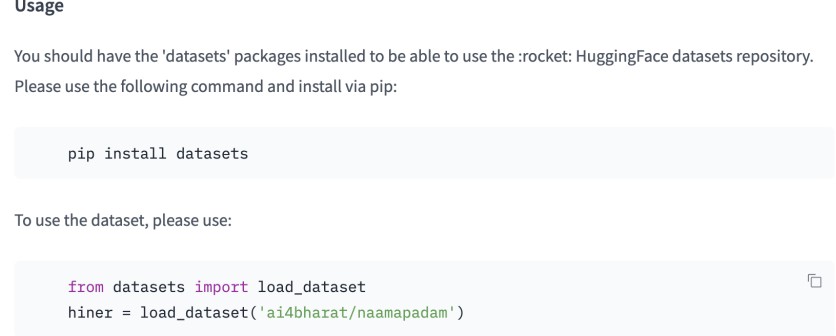

Figure S2:  **Example of a *Usage* Section**

Intuitively, a *Usage* section could give users quick instructions on how to use the dataset, which could make the dataset more accessible, transparent, and reproducible. To verify this intuition, we conduct an experiment to quantify how the *Usage* section will affect the dataset's popularity.

In our experiment, we trained a BERT (Devlin et al., 2018) Model using the content of dataset cards and their corresponding download counts. To ensure comparability, the download counts were normalized to a range of [0,1] and stratified monthly based on the dataset's creation time. This ranking system assigned a rank of 1 to the dataset with the highest downloads within a given month, and a rank of 0 to the dataset with the lowest downloads.

Using the dataset card content, the trained BERT Model predicted the download counts. Subsequently, we conducted a test using 567 dataset cards that included a *Usage* section. For this test, we deliberately removed the *Usage* section from the dataset cards and employed the BERT Model to predict the download counts for these modified cards. The resulting predictions are summarized in **Table. S3**. The average predicted score of downloads after removing the *Usage* section is 0.0185 lower compared to the original dataset card. This indicates a decrease in the number of downloads, highlighting the negative impact of not including a *Usage* section.

In future research, it would be valuable to further investigate the effect of adding a *Usage* section to the dataset cards that do not have one originally. A randomized controlled trial (RCT) experiment could be conducted to assess whether the inclusion of a *Usage* section leads to an increase in downloads.

| Condition | Predicted Score of Downloads |
|---|---|
| With *Usage* Section | 0.3917 |
| Without *Usage* Section | 0.3732 |
| Change in Score | -0.0185 |

Table S3: **Predicted Impact of *Usage* Section on Dataset Downloads.** This table presents a comparative analysis of predicted download scores for dataset cards, distinguishing between those that include a *Usage* Section and those from which it has been removed. It indicates a potential decrease in download rates following the removal of the *Usage* Section.

## D    OPTIONAL METRICS FOR DATASETS

In our analysis, we employ downloads as a metric to gauge the popularity of the dataset. Numerous factors can influence the download count, including the dataset's publication date and its associated research field. Moreover, aside from dataset downloads, we can incorporate other indicators of dataset popularity, such as the count of models utilizing the datasets and the corresponding download counts.

To address the concerns of factors that might affect downloads, we expanded our dataset analysis by extracting more metadata from the Hugging Face dataset information. We collected data such as the models utilizing the corresponding dataset, the total number of downloads for these models, and the dataset's task domain. The primary dataset tasks recognized by Hugging Face encompass Multimodal, Computer Vision, Natural Language Processing, Audio, Tabular and Reinforcement Learning. Among the total of 7,433 dataset cards, 1,988 are categorized as NLP dataset cards, 198 are related to computer vision, and 102 pertain to multimodal datasets. We proceeded with additional analysis by employing the following metrics:

1. We integrated dataset downloads (*"direct usage"*) with the downloads of models employing the dataset (*"secondary usage"*).

2. A time range (measured in months) was selected, encompassing dataset cards created within the designated time frame and specified task domain.

3. Selected dataset cards were ranked within each domain for each time range and then normalized to a range of $[0, 1]$.

By adopting this approach, we were able to compare dataset cards created in the same month and task domain, assessing them based on the metrics of direct and secondary usage metrics. We conducted a word count analysis using this new metric and attained results consistent with our prior analysis that datasets with higher rankings tend to have longer dataset cards, as shown in **Fig. S3**.

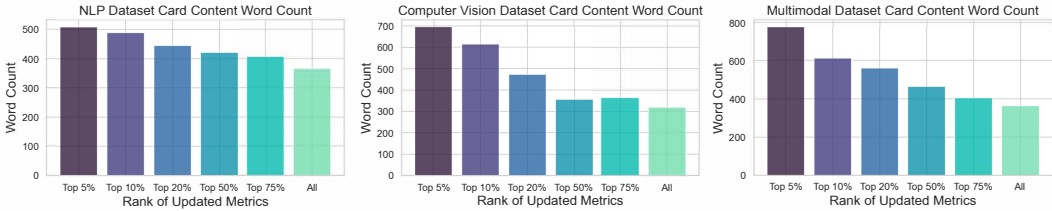

Figure S3: **Word Count Variation Based on Direct and Secondary Usage Rankings.** This figure demonstrates the relationship between the length of dataset cards and their rankings in terms of direct and secondary usage. It reveals a distinct pattern: dataset cards with higher rankings tend to have a greater word count, suggesting a correlation with more thorough and detailed content.

The finding enables us to contemplate an alternative metric option, factoring in publication time, research area, and secondary dataset usage. However, the results remain aligned with our previous analysis, which solely considered download counts, highlighting the reasonableness of using download counts as metrics.

## E  APPLICABILITY ACROSS PLATFORMS: ADAPTING TO GITHUB

Our study demonstrates strong potential for application across various platforms. The foundational format of Hugging Face's dataset cards, essentially README files, is a prevalent documentation standard shared by many platforms, notably GitHub. This commonality implies that our approach to parsing and analyzing dataset cards can be readily adapted for broader studies. To illustrate, we present an example of how our analysis methodology can be effectively applied to GitHub, a widely recognized open-source platform for data and code sharing.

Our expanded analysis involved sourcing datasets from a GitHub repository of Papers With Code[1]. We chose repositories linked to dataset-relevant papers and processed their README files using the pipeline proposed in our paper on Hugging Face dataset card analysis. This exploration revealed a more varied structure in GitHub's dataset cards. For example, 57% of the section titles on GitHub are unique, compared to just 3% on Hugging Face. Due to their specificity, we excluded these unique sections and created a categorization list based on Hugging Face's community-endorsed dataset card structure, mapping GitHub's titles through keyword matching. This method successfully categorized 74% of GitHub's section titles.

As shown in **Table. S4**, our analysis reveals that both platforms excel in *Dataset Description* and *Additional Information* sections but underperform in *Dataset Creation* and *Considerations for Using the Data*, underscoring points raised in our paper. A notable difference is GitHub's lower emphasis on *Dataset Structure*, highlighting the potentially positive impact of Hugging Face's community-endorsed dataset structure. Furthermore, the prevalence of *Usage* and *Experiment* sections on GitHub, absent in Hugging Face, highlights the practical value of these sections in promoting the usability of datasets. Adopting these sections, as suggested in our paper, could enrich the structure of Hugging Face's dataset cards, making them more comprehensive and practically useful.

These results indicate our method's adaptability to other platforms and provide a benchmark for evaluating dataset documentation elsewhere. The insights from our Hugging Face study can guide the categorization and enhancement of dataset documentation across various platforms, especially in the current situation that most other platforms don't have a standardized dataset card structure.

| Section Type | GitHub | Hugging Face | Description |
|---|---|---|---|
| Dataset Description | 0.62 | 0.46 | Summary, leaderboard, languages, etc. |
| Dataset Structure | 0.09 | 0.34 | Format, fields, splits, etc. |
| Dataset Creation | 0.08 | 0.15 | Motivation, collection procedures, etc. |
| Considerations for Using the Data | 0.02 | 0.08 | Limitations, biases, disclaimers, etc. |
| Additional Information | 0.62 | 0.58 | Citations, acknowledgements, licensing, etc. |
| Experiment | 0.57 | - | Model experiments, training, evaluation on the dataset, etc. |
| Usage | 0.38 | - | Instructions for setup, installation, requirements, etc. |

Table S4: **Comparison of Fill-out Rate of Dataset Documentation on GitHub and Hugging Face.** Dataset cards from both GitHub and Hugging Face perform well in the *Dataset Description* and *Additional Information* sections, but fall short in the *Dataset Creation* and *Considerations for Using the Data* sections. While GitHub places less emphasis on *Dataset Structure*, it shows a higher occurrence of *Usage* and *Experiment* sections.

---

[1]https://github.com/paperswithcode/paperswithcode-data

# F ADDITIONAL FIGURES AND TABLES

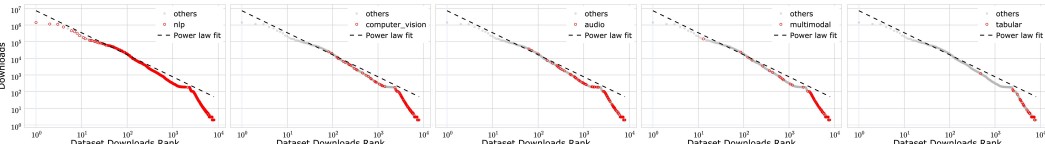

Figure S4: **Power Law Distribution Patterns in Dataset Usage across Task Domains.** This figure illustrates the dataset usage distribution within each task domain, demonstrating a consistent power law distribution, despite the variations in the number of datasets across different domains.

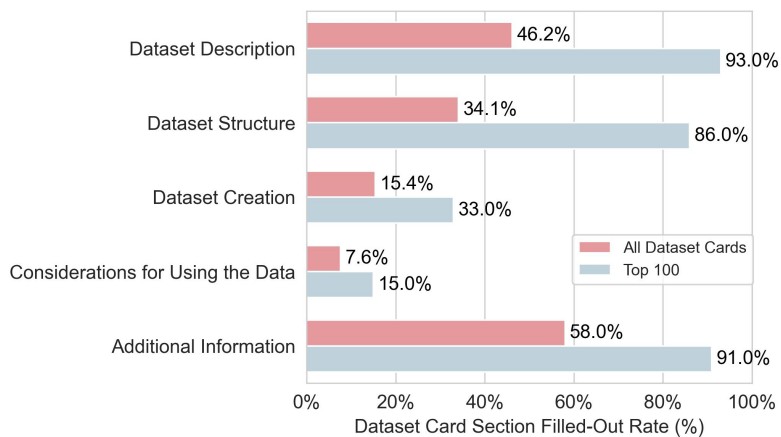

Figure S5: **Highly Downloaded Dataset Cards Exhibit Greater Completion across All Sections.** This figure indicates that the top 100 downloaded dataset cards exhibit a higher completion rate compared to all dataset cards in the sections recommended by the Hugging Face community. However, there is a consistently low completion rate in the *Dataset Creation* and *Considerations for Using the Data sections*, regardless of the dataset cards' popularity.

| Category | Description | Dataset Card Number | Adherence to Guidelines | Avg. Word Count |
|---|---|---|---|---|
| Industry organization | Companies (e.g. Hugging Face, Facebook) | 2,527 | **0.34** | 219 |
| Academic organization | Universities, Research Labs (e.g. Stanford CRFM, jhu-clsp) | 985 | 0.31 | **427** |
| Community | Non-profit Communities (e.g. allenai, bio-datasets) | 1,387 | 0.27 | 190 |
| Industry professional | Engineers, Industry Scientists | 985 | 0.25 | 256 |
| Academic professional | Students, Postdocs, Faculty | 672 | 0.16 | 180 |
| All dataset cards | 7,433 dataset cards analyzed | 7,433 | 0.29 | 234 |

Table S5: **Differences in the Practices of Dataset Documentation across Creators from Different Backgrounds.** This table highlights the diverse documentation practices across creators from different backgrounds. Industry organizations, with the most creators, adhere to the guidelines best. Academics, though fewer, offer the most comprehensive documentation, while academic professionals exhibit lower guideline adherence and shorter word counts. The information about these creators is gathered from their linked GitHub, Twitter, and personal websites on their Hugging Face profiles.

