# OpenReview forum: "Navigating Dataset Documentations in AI: A Large-Scale Analysis of Dataset Cards on HuggingFace"
_ICLR.cc/2024/Conference — ICLR 2024 poster_

### Official Review · Reviewer_ALk3 · 2023-10-30

**Soundness:** 2 fair
**Presentation:** 2 fair
**Contribution:** 2 fair
**Rating:** 5
**Confidence:** 3

**Summary:**

The paper conducts a large scale analysis on dataset cards from HuggingFace. The paper contributes mainly 5 findings which offers a unique perspective on dataset documentations.

**Strengths:**

1. The paper is well written with each finding clearly explained and backed by statistical data analysis
2. Through various data analysis, the paper showed quite a few insights on the current status of dataset cards, which are often under-studied from previous works.
3. The paper empirically verified the hypothesis proposed by employing human annotations.
4. Through the study of HuggingFace dataset cards, the paper proposes new documentations practices that could be adopted by the industry to improve dataset documentations.

**Weaknesses:**

1. The paper only analyzes the documentations from HuggingFace, thus the potential for the findings to apply to other platforms remains unclear. Are there any suggestions from the authors on how to process dataset cards on other platforms?
2. The paper uses exact keyword matching to identify corresponding subsections. Thus it's hard to know the proportion of dataset cards which covers the corresponding subsection but with different keywords.
3. Although the paper proposes new standard that the industry should follow, are there any methods the authors think that could help achieve this goal?
4. How are the human annotators selected? Are they machine learning domain experts?

**Questions:**

See my questions in weakness section

---

> ### Author Response · Authors · 2023-11-18
> **Response to Reviewer ALk3 (1/2)**
>
> We appreciate reviewer ALk3 for their thoughtful feedback and the recognition of its strengths in our work. We address many of reviewer ALk3's comments in our general response above, and we offer more specific details on certain comments below.
>
> **Applicability of Findings to Other Platforms**
>
> > *W1. "The paper only analyzes the documentations from HuggingFace, thus the potential for the findings to apply to other platforms remains unclear. Are there any suggestions from the authors on how to process dataset cards on other platforms?"*
>
> In response to reviewer ALk3's concern about the applicability of our findings to other platforms, we would like to emphasize the following points:
>
> - **Representative Case Study:** Hugging Face is one of the most popular ML platforms, which can serve as a representative case study. The trends and practices observed here are likely to be indicative of broader patterns in dataset documentation across various platforms, given the platform's influence and widespread adoption in the ML community.
>
> -  **Transferability of Methodology:** The fundamental structure of Hugging Face's dataset cards, which are README files, is a common documentation framework used widely across platforms, including GitHub. This similarity suggests that our methodology of parsing and analyzing the dataset cards can be easily transferred to other platforms. We have made available the code for data retrieval and analysis, as well as the dataset, at https://anonymous.4open.science/r/HuggingFace-Dataset-Card-Analysis, to facilitate further research.
>
> **Validation of Exact Keyword Matching**
>
> > *W2. "The paper uses exact keyword matching to identify corresponding subsections. Thus it's hard to know the proportion of dataset cards which cover the corresponding subsection but with different keywords."*
>
> Notably, **84%** of the occurrences of section titles in the 7,433 dataset cards align with the 27 titles provided by the HuggingFace community. This high level of adherence demonstrates the effectiveness of exact keyword matching as an analytical method. Additionally, the fact that only 11% of dataset cards do not include any of these 27 standardized titles further strengthens the argument. This low rate of deviation indicates that the majority of dataset cards follow a consistent structure, underscoring the reliability of exact keyword matching in identifying corresponding dataset card sections.
>
> We have added this validation procedure to *Appendix C.1 Accessing and Parsing Dataset Cards* to clarify this point.
>
> **Proposals for ImprovinDocumentation Standards**
>
> > *W3. "Although the paper proposes new standard that the industry should follow, are there any methods the authors think that could help achieve this goal?"*
>
> We have identified **two** key methods that could help achieve this goal:
>
> 1. **Standardization of Dataset Documentation**
>
> Based on our findings, we emphasize the high adherence to the HuggingFace community-endorsed dataset card structure among high-downloaded dataset cards and the general high adherence to the structure of subsections within sections provided by the HuggingFace community-endorsed dataset card. This suggests that an official template for dataset documentation, such as the one provided by HuggingFace, is an effective approach for encouraging practitioners to incorporate essential information within their dataset documentation. In this sense, it is also important for the community to work together to develop a standard for dataset documentation. For example, we found that the section "Uses" is missing in the HuggingFace community-endorsed dataset card, but is important as implied by our empirical analysis and identified in a previous study [1].
>
> 2. **Employing Automatic Tools**
>
>  Hugging Face offers the convenience of importing a dataset card template link to automatically generate a template with all the necessary fields pre-filled. Such a function is a good example of how automatic tools can be used to facilitate the creation of dataset documentation. Notably, the community's high compliance suggests the efficacy of such automated features. Several startups, including ValidMind, are also actively developing automatic tools for dataset documentation, contributing to the evolving landscape of efficient documentation practices for both models and data.
>
>  Beyond the automatic generation of dataset documentation, the validation of such documentation holds equal significance. We propose that the dataset community explore the integration of automatic validation mechanisms within dataset hosting platforms. These platforms could introduce a structured checklist for dataset documentation, with the documentation subject to automated validation based on this checklist. This approach not only incentivizes practitioners to produce more comprehensive and high-quality dataset documentation but also fosters improvements in dataset usability, transparency, and reproducibility.

---

> ### Author Response · Authors · 2023-11-18
> **Response to Reviewer ALk3 (2/2)**
>
> **Selection of Human Annotators**
>
>
> > *W4. "How are the human annotators selected? Are they machine learning domain experts?"*
>
>
> In our study, the human annotators were carefully chosen, taking into consideration their expertise and background in the field of AI. Specifically, we selected PhD students with a robust research background in NLP, Computer Vision, Human-AI, Machine Learning, and Data Science. Their extensive experience with AI and utilizing datasets ensured a deep understanding of dataset documentation. To confirm the reliability of our evaluation, we randomly sampled 30 dataset cards for the annotators to assess and achieved an Intraclass Correlation Coefficient (ICC) of **0.76**, which is considered a good agreement [2]. This high level of agreement, combined with the annotators' deep expertise in AI research, substantially reinforces the trustworthiness of the annotation results.
>
> We have added the selection criteria of human annotators to *Appendix C.2 Human-Annotated Dataset Card Evaluation Methodology and Criteria*.
>
> **Reference**
>
> [1] Gebru, Timnit, et al. "Datasheets for datasets." Communications of the ACM 64.12 (2021): 86-92.
>
> [2] Koo, Terry K., and Mae Y. Li. "A guideline of selecting and reporting intraclass correlation coefficients for reliability research." Journal of chiropractic medicine 15.2 (2016): 155-163.

---

> > ### Author Response · Authors · 2023-11-21
> > **We would like to hear back from Reviewer ALk3**
> >
> > Dear Reviewer ALk3,
> >
> > We would like to follow up to see if our response and revised manuscript address your concerns or if you have any further questions. Should there remain any points that require further attention, we would be grateful for the chance to engage in further discussion.
> >
> > Thank you for your time and consideration.
> >
> > Best regards,
> >
> > Authors

---

> > > ### Comment · Reviewer_ALk3 · 2023-11-22
> > > **Thank the authors for the response**
> > >
> > > Thank the authors for the responses. It clears some of my doubts. However, I still feel the method is too limited to the HuggingFace community. Perhaps the proposed method can generalize to other platforms as well, but it's not studied in-depth in the paper.
> > >
> > > Furthermore, the benefit of this work on this research direction is also limited. Since the paper only focuses on HuggingFace, the pipeline proposed in the paper can just be run once and then it's done just like the writing of this paper. I think including more study on how these problem can be solved and show proof of it in the paper itself just like the responses the authors gave in W3 will greatly increase the integrality of this paper.
> > >
> > > Overall, I feel this paper is well written and the analysis is thorough. It shows some interesting findings of the data cards but with limited applicability and lacks the study of how these findings/problems can be best utilized or solved. Therefore I will keep my original score.

---

> ### Author Response · Authors · 2023-11-23
> **2nd Round of Response to Reviewer ALk3 (1/2)**
>
> We thank Reviewer ALk3 for their response, especially their recognition of the thoroughness of our analysis and the interesting findings we uncovered. To further clarify and expand upon the broader implications of our findings, we have conducted additional analysis and discussions.
>
>
> **(1) Reproducing Our Analysis: Case Study on GitHub**
>
> Our expanded analysis involved sourcing datasets from a GitHub repository of [Papers With Code](https://github.com/paperswithcode/paperswithcode-data).  We identified 3,182 dataset repositories linked to dataset-relevant papers and then processed their README files using the pipeline proposed in our paper on Hugging Face dataset card analysis. This exploration revealed a more varied structure in GitHub's dataset cards. For example, 57% of the section titles on GitHub are unique, compared to just 3% on Hugging Face. Due to their specificity, we excluded these unique sections and created a categorization list based on Hugging Face's community-endorsed dataset card structure, mapping GitHub's titles through keyword matching. This method successfully categorized 74% of GitHub's section titles. The table below presents the completion rates for each type of section in the GitHub and Hugging Face dataset cards (refer to Table S4 in our revised manuscript).
>
> | Section Type                      | GitHub | Hugging Face | Description                                              |
> | --------------------------------- | ------ | ------------ | -------------------------------------------------------- |
> | Dataset Description               | 0.62   | 0.46         | Summary, leaderboard, languages, etc.                    |
> | Dataset Structure                 | 0.09   | 0.34         | Format, fields, splits, etc.                             |
> | Dataset Creation                  | 0.08   | 0.15         | Motivation, collection procedures, etc.                  |
> | Considerations for Using the Data | 0.02   | 0.08         | Limitations, biases, disclaimers, etc.                   |
> | Additional Information            | 0.62   | 0.58         | Citations, acknowledgements, licensing, etc.             |
> | Experiment                        | 0.57   | -            | Model experiments, training, evaluation on the dataset, etc. |
> | Usage                             | 0.38   | -            | Instructions for setup, installation, requirements, etc. |
>
> Our analysis reveals that both platforms excel in the "Dataset Description" and "Additional Information" sections but underperform in "Dataset Creation" and "Considerations for Using the Data", underscoring points raised in our paper. A notable difference is GitHub's lower emphasis on "Dataset Structure", highlighting the potentially positive impact of Hugging Face's community-endorsed dataset structure. Furthermore, the prevalence of the "Usage" and "Experiment" sections on GitHub, absent in Hugging Face, underscores the practical value of these sections in promoting the usability of datasets. Adopting these sections, as suggested in our paper, could enrich the structure of Hugging Face's dataset cards, making them more comprehensive and practically useful.
>
> These results indicate our method's adaptability to other platforms and provide a benchmark for evaluating dataset documentation elsewhere. The insights from our Hugging Face study can guide the categorization and enhancement of dataset documentation across various platforms, especially in the current situation that most other platforms don't have a standardized dataset card structure.

---

> ### Author Response · Authors · 2023-11-23
> **2nd Round of Response to Reviewer ALk3 (2/2)**
>
> **(2) Significance of the Hugging Face Platform**
>
> The increasing popularity of the Hugging Face platform within the machine learning (ML) community positions it as an exemplary subject for empirical studies, potentially influencing dataset documentation practices across various platforms. Its growing influence and widespread adoption make it a representative case study, with trends and practices observed on Hugging Face likely reflective of broader patterns in the ML field. The platform's user-friendly interface, coupled with automated features like the dataset card template, further solidifies its suitability for in-depth analysis of ML practices.
>
> Moreover, Hugging Face has already been the focus of various academic studies, underscoring its relevance in the research community. For instance, [1] explored the carbon footprint of ML models hosted on Hugging Face, while [2] and [3] investigated model evaluation, reusability, and best practices. These studies validate the platform's significance in advancing our understanding of ML trends and practices.
>
> Therefore, examining the Hugging Face community is a strategic starting point for broader investigations into dataset documentation across multiple platforms. The insights gained from such a focused study can guide and inform similar analyses on other platforms, contributing to the overall advancement of ML practices.
>
> We have accordingly revised our manuscript in *Discussion* and *Appendix G Applicability across Platforms: Adapting to GitHub* sections, to emphasize the transferability of our study's methodology to other platforms. We hope our response could address your concerns about the applicability of our findings to other platforms.  There could be more interesting findings to explore in the future, and we hope our work can serve as a good starting point and provide some references for future studies.
>
> [1] Castaño, Joel, et al. "Exploring the Carbon Footprint of Hugging Face's ML Models: A Repository Mining Study." arXiv preprint arXiv:2305.11164 (2023).
>
> [2] Von Werra, Leandro, et al. "Evaluate & Evaluation on the Hub: Better Best Practices for Data and Model Measurements." Proceedings of the 2022 Conference on Empirical Methods in Natural Language Processing: System Demonstrations. 2022.
>
> [3] Jiang, Wenxin, et al. "An empirical study of pre-trained model reuse in the hugging face deep learning model registry." arXiv preprint arXiv:2303.02552 (2023).

---

### Official Review · Reviewer_kkaF · 2023-11-01

**Soundness:** 3 good
**Presentation:** 3 good
**Contribution:** 3 good
**Rating:** 8
**Confidence:** 4

**Summary:**

This paper gives an analysis of dataset documentation practices by data creators. They focus on a Hugging Face’s data repository. The authors highlight findings such as frequently downloaded datasets are also often well documented, the fact that social concerns are some of the least documented and that users care about operational matters related to dataset - these may include, versioning, download details… etc. This work is a continuation of work championed by many ML researchers concerned with documentation and auditing to better understand ethical implications of our tools.

**Strengths:**

It is useful to provide these kinds of evaluations on new documentation processes. The ideas of how to document the different pieces of the ML process, though important, are still new and should change where there is a need. This type of study gives us a way to see where things are working and where improvements are needed. Eg. Inclusion of a uses section. Additionally, the authors started hinting on some of the relationships between what sections of the data card get filled in and the motivations of the data creators. Additionally,it is once again reminded  that we have to continue searching for better incentives to get dataset creators better document their datasets beyond describing what is in there. This work does not give us these answers and they are perhaps beyond the scope of this work but it is certainly another nudge that there is a lot more work here.

**Weaknesses:**

None for me. I think the paper did what it promised to do.

**Questions:**

- Were the authors able to find a correlation between well documented datasets and their creators? Eg, Do they come from other disciplines? Are they academic or industry researchers? Start-ups...etc
- Is there a relationship between the features included in the dataset and whether there is going to be a limitations section? ie do datasets that contain more sensitive data likely to be well documented and include limitations sections?

---

> ### Author Response · Authors · 2023-11-18
> **Response to Reviewer kkaF (1/2)**
>
> We thank reviewer kkaF for their positive assessment of our paper and for recognizing the importance of our work. We appreciate their support and would like to address the queries.
>
>
> > *Q1. "Were the authors able to find a correlation between well documented datasets and their creators? Eg, Do they come from other disciplines? Are they academic or industry researchers? Start-ups...etc"*
>
>
> Our analysis of 1,982 distinct creators who contribute to the 7,433 dataset cards, reveals significant variations in dataset documentation practices based on the creators' backgrounds. The table below summarizes the results of our analysis (we have added this table as Table S3 in Appendix F of our manuscript):
>
>
> | Category              | Description                                                                                                                              | Dataset Number | Adherence to Guidelines | Avg. Word Count |
> | --------------------- | ---------------------------------------------------------------------------------------------------------------------------------------- | -------------- | ----------------------- | --------------- |
> | Industry organization | Companies (e.g.[Hugging Face](https://huggingface.co/huggingface), [facebook](https://huggingface.co/facebook))                                 | 2527           | **0.34**          | 219             |
> | Academic organization | Universities, Research Labs (e.g.[Stanford CRFM](https://huggingface.co/stanford-crfm), [jhu-clsp](https://huggingface.co/jhu-clsp))           | 985            | 0.31                    | **427**   |
> | Community             | Non-profit Communities (e.g.[bio-datasets](https://huggingface.co/bio-datasets), [keras-dreambooth](https://huggingface.co/keras-dreambooth)) | 1387           | 0.27                    | 190             |
> | Industry professional | Engineers, Industry Scientists                                                                                                           | 985            | 0.25                    | 256             |
> | Academic professional | Students, Postdocs, Faculty                                                                                                              | 672            | 0.16                    | 180             |
> | All dataset card      |                                                                                                                                          | 7433           | 0.29                    | 234             |
>
>
> Industry organizations, contributing 34% of the datasets and 90% of the top 100 downloads, lead in adherence to guidelines with an average word count of 219, indicating a commitment to detailed documentation. Academic organizations, though fewer in number, showcase comprehensive documentation with the highest average word count of 427. Academic professionals, however, show lower adherence to guidelines and a reduced word count, pointing to a varied relationship between the creator's sector and their approach to dataset documentation.
>
>
> The information about these creators is gathered from their linked GitHub, Twitter, and personal websites on their Hugging Face profiles. Generally, datasets produced by organizations tend to be of higher quality compared to those created by individuals. Additionally, industry-generated dataset cards often surpass those from academic sources in quality.

---

> ### Author Response · Authors · 2023-11-18
> **Response to Reviewer kkaF (2/2)**
>
> > *Q2. Is there a relationship between the features included in the dataset and whether there is going to be a limitations section? ie do datasets that contain more sensitive data likely to be well documented and include limitations sections?*
>
> We thank Reviewer kkaF's intriguing question about the relationship between dataset sensitivity and documentation quality. We delved into the features of the dataset and found about 36% dataset may contain sensitive data among the 577 dataset cards that have the "Considerations for Using the Data" section.
>
> | Category     | Number | Example                                                                                                                                                                                                                                                                                  |
> | ------------ | ------ | ---------------------------------------------------------------------------------------------------------------------------------------------------------------------------------------------------------------------------------------------------------------------------------------- |
> | Website Data | 158    | (1) "Since our dataset contains tables that are scraped from the web, it will also contain many toxic, racist, sexist, and otherwise harmful biases and texts." (2) "We are aware that since the data comes from public web pages, some biases may be present in the dataset."              |
> | News Data    | 62     | (1)"News articles have been shown to conform to writing conventions" (2)"As one can imagine, data contains contemporary public figures or individuals who appeared in the news."                                                                                                         |
> | Medical Data | 46     | (1)"A handful of disease concepts were discovered that were not included in MEDIC." (2)"This dataset of protein interactions was manually curated by experts utilizing published scientific literature."                                                                                  |
> | Voice Data   | 11     | (1)"The Mongolian and Ukrainian languages are spelled as "Mangolian" and "Ukranian" in this version of the dataset." (2)"The dataset consists of people who have donated their voice online. You agree to not attempt to determine the identity of speakers in the Common Voice dataset." |
>
>
> It is notable that as discussed in Section 5 of our paper, there are various topics being discussed in the "Considerations fo Using the Data" section, like the technical limitations and biases. Some dataset curators might focus solely on technical biases, neglecting ethical aspects (e.g., [allenai/nllb](https://huggingface.co/datasets/allenai/nllb#considerations-for-using-the-data)). Moreover, the proportion of datasets containing sensitive data that include the "Considerations for Using the Data" section is also low. For instance, a mere 9.2% of medical datasets incorporate this critical consideration. Coupled with our discovery that the "Considerations for Using the Data" section constitutes only 2.1% of the total text, this underscores a substantial need for improvement within this domain.
>
> In response to your encouraging comments, we are motivated to continue our work and uncover more insights about dataset documentation.

---

### Official Review · Reviewer_eSEd · 2023-11-06

**Soundness:** 2 fair
**Presentation:** 4 excellent
**Contribution:** 2 fair
**Rating:** 5
**Confidence:** 4

**Summary:**

Authors conduct an analysis on a large array of Hugging face datasets, and their associated documentation. Their aim is to better understand what results in the popularity of a particular datasets, and perhaps standardize the documentation procedure of datasets

**Strengths:**

The paper is well written, easy to follow and digest. It is also interesting to study a large array of datasets.

**Weaknesses:**

- Most finding in the paper are “common sense” i.e. there always has been a high correlation with quality of documentation and usability of a particular tool/dataset
- It’s not clear how this paper fits ICLR conference, although it the study does conduct a comprehensive overview of the hugging face dataset cards and provides an insight on what makes the datasets popular they aren’t significantly new and their insight is limited
   - For example datasets may be simply popular because they are in a “hot” area of study
   - Another example is the quality of the underlying dataset – it may be the case the reason one of the datasets is very popular is simply attributed to the fact that there are many studies leveraging this dataset (which in some sense is correlated with the quality of the data)
- It seems a although there are several findings with respect to what makes a dataset popular, there isn’t a conclusive suggestion for improving dataset sharing/contribution.

**Questions:**

- The correlation between documentation length and popularity is not causal
   - I’d emphasize on the quality of the documentation and what points these “popular cards” address in the written documentation there may be a more granular set of points mentioned in each that results in their popularity
        - Data usability looks to have been a common key topic
- It’s not clear how the power law with respect to data usage is useful
   - Power-law is quite pervasive -- for example in “popularity”
   - This power-law may simply represent the underlying distribution of people interested in deep learning topics/domains
      - E.g. text (NLP) versus mortgage tabular data (Finance)
- There’s a key component in the dataset documentation that is missing e.g. how the data is processed – perhaps this is covered in data usability?
   - By having a raw data card with associated derivatives simply being a modified transformation of the raw could drastically help with reproducibility
   - Further this way of representing allows for more comprehensive analysis on how various preprocessing steps effect benchmarks
       - these are much more important considerations in time-series, and healthcare domains for example.

---

> ### Author Response · Authors · 2023-11-18
> **Response to Reviewer eSEd (1/3)**
>
> Thank you very much for your time and feedback. We really appreciate it.
>
> **"Common Sense" Findings**
>
> > *"Most finding in the paper are "common sense""*
>
> While some of our findings may seem intuitive, our study uniquely provides **empirical** backing for these insights, which has not been systematically shown before. The link between documentation quality and dataset usability, while anticipated, has not been empirically confirmed on such a large-scale dataset until now. Our comprehensive analysis of 7,433 dataset cards from the Hugging Face platform, a leading ML platform, offers robust statistical support, moving our understanding from theoretical to empirical.
> For instance, our study uncovers a notable lack of emphasis on crucial sections like "Dataset Creation" and "Considerations for Using the Data," which make up just 10% and 2.1% of the total text in dataset cards, respectively. These insights are not only novel but also practical, highlighting significant opportunities for enhancing dataset documentation practices.
>
> Moreover, the scalability and reproducibility of our analysis methodology are noteworthy. It can be readily applied to other platforms, like GitHub, which similarly utilizes README files for dataset documentation. This study represents a pivotal step in conducting extensive empirical analysis of dataset documentation practices, with a focus on Hugging Face as a leading case study in the ML community. We hope to uncover and examine the prevalent norms and methods currently employed in dataset documentation. By doing so, we aim to not only underscore the vital importance of ongoing improvements in dataset documentation practices but also contribute to the responsible and ethical deployment of AI community.
>
> **Relevance to ICLR Conference**
>
> > *"It’s not clear how this paper fits ICLR conference"*
>
> - **Data-Centric AI Research is a core part of ML:** The evolving focus of the Machine Learning community on Data-centric AI is evident in the recent trends at major conferences, including ICLR. Significantly, ICLR has introduced **"datasets and benchmarks"** as an explicit topic this year, reflecting its growing interest in this domain. Previous ICLR conferences have accepted studies analyzing dataset practices, such as [1] and [2]. Furthermore, NeurIPS, another leading ML conference, started a separate Datasets and Benchmarks Track in 2021. Notably, [3] conducted a comprehensive dataset analysis and was recognized with a best paper award. While there is an abundance of research on analyzing and creating datasets, our work uniquely addresses the under-explored area of dataset documentation practices, underscoring its relevance and importance to the ML community and ICLR.
> - **Importance of Dataset Documentation in ML research:**
>  Our study directly contributes to the field of machine learning by offering a detailed analysis of dataset documentation practices, a vital yet often overlooked aspect of ML research. The insights and findings from our research have practical implications, extending beyond academic interest. By identifying gaps in dataset documentation and suggesting improvements, our study provides actionable recommendations for practitioners and the community. For instance, we proposed the addition of a usage section in dataset card templates to the Hugging Face community. This recommendation aligns with ICLR's focus on impactful research that advances the field of ML, emphasizing the significance and suitability of our study for the conference.
>
> **Insights of Popular Datasets**
>
> > *"Datas may be simply popular because they are in a “hot” area of study. Another example is the quality of the underlying dataset"*
>
> The correlation between dataset popularity and documentation quality can **inform better dataset card design and practice**. While popularity may stem from being in a 'hot' area or the underlying quality of the dataset and can provide insights into what current research is focusing on and predict the future hot topic, our study offers a more nuanced perspective, exploring the relationship between dataset documentation and their popularity. This approach reveals distinct documentation patterns between popular and less popular datasets, enabling us to **identify good practices and common areas for improvement**. For instance, we observed that popular datasets often adhere better to the Hugging Face community-endorsed structure for dataset cards and exhibit higher word counts. These characteristics can serve as practical guidelines for dataset creators looking to improve their dataset cards. Additionally, we identified a consistent lack of emphasis on the "Dataset Creation" and "Considerations for Using the Data" sections across both popular and less popular datasets. This finding highlights the need for the community to create more compelling incentives, encouraging dataset creators to enhance documentation, particularly for these critical sections that are often neglected.

---

> ### Author Response · Authors · 2023-11-18
> **Response to Reviewer eSEd (2/3)**
>
> **Suggestion for Improving Dataset Sharing/Contribution**
>
>
> > *"It seems a although there are several findings with respect to what makes a dataset popular, there isn’t a conclusive suggestion for improving dataset sharing/contribution."*
>
>
> We offer recommendations for improving dataset documentation for better dataset sharing, drawing from key findings from our research:
>
>
> - **Adherence to Community-Endorsed Dataset Card Structure**: Our analysis indicates that highly downloaded dataset cards often comply with the Hugging Face community-endorsed structure. We advocate for dataset creators to consistently adhere to this structure, enhancing standardization and clarity.
> - **Emphasis on Critical Sections:** We advise dataset creators to focus more on the "Dataset Creation" and "Considerations for Using the Data" sections, which are the least detailed in the current documentation. These sections are vital for dataset reproducibility and usability, and addressing them properly can mitigate potential limitations and ethical concerns.
> - **Content Suggestions for Each Section:** We delve into the specifics of each section and subsection, providing suggestions for practitioners to write the sections. For instance, in the "Considerations for Using the Data" section, they can discuss both technical and social aspects of dataset limitations and impacts.
> - **Inclusion of a 'Usage' Section:** Despite its absence in the Hugging Face community-endorsed template, our analysis reveals that a "Usage" section is frequently included in dataset cards and significantly enhances dataset transparency and usability. We recommend the community incorporate this section into the standard template.
> - **Humain Evaluation of Dataset Card Quality:**  Through human evaluation, we discovered a strong correlation between the comprehensiveness of dataset cards and their overall quality. We suggest that creators improve thoroughness in their documentation.
>
>
> Overall, improving the dataset documentation can help improve the reproducibility, transparency, and accessibility of the dataset, which can help promote the dataset sharing and contribution ecosystem. These suggestions are implied in our manuscript, encompassing sections 4 to 6, as well as in the Discussion section.
>
>
> **The Correlation Between Dataset Cards and Dataset Popularity**
>
>
> > *"Q1. The correlation between documentation length and popularity is not causal."*
> >
> > * *"I’d emphasize on the quality of the documentation and what points these “popular cards” address in the written documentation there may be a more granular set of points mentioned in each that results in their popularity**"*
>
>
> The analysis between the dataset documentation practices and the dataset popularity reveals a strongly positive relationship, in aspects of adherence to the template, word count, and the fill-out rate within each section and subsection. The main findings include:
>
>
> - **Better Adherence to Community-recommended Dataset Card:** Popular datasets exhibit better adherence to the community-recommended documentation structure.
> - **Longer Word Length:** Popular datasets feature a higher word count, particularly notable in the "Dataset Description" section.
> - **Higher Fill-Out Rate across Dataset Structure:** There's a higher completion rate across all documentation sections in these datasets, with a notable emphasis on the Dataset Structure section. A well-documented structure enhances the understanding and usability of the dataset.
>
>
> The table below illustrates the completion rates in different sections of the documentation (we have added this result as Figure S6 in Appendix F in our manuscript):
>
>
> |                   | Dataset Description | Dataset Structure | Dataset Creation | Considerations for Using the Data | Additional Information |
> | ----------------- | ------------------- | ----------------- | ---------------- | --------------------------------- | ---------------------- |
> | Top 100           | 0.93                | 0.86              | 0.34             | 0.15                              | 0.91                   |
> | All dataset cards | 0.47                | 0.35              | 0.16             | 0.08                              | 0.61                   |
>
>
> By examining the correlation between documentation practices and dataset popularity, we gain insights into effective documentation strategies. The strengths observed in popular datasets can guide other creators, serving as a template for effective documentation. Notably, the lower completion rates in sections like "Considerations for Using the Data" in even the most downloaded datasets highlight areas needing more focus. These findings can also serve as a benchmark for the community to establish standards in dataset documentation.

---

> > ### Author Response · Authors · 2023-11-21
> > **We would like to hear back from Reviewer eSEd**
> >
> > Dear Reviewer eSEd,
> >
> > We hope this message finds you well. We wanted to check in and see if our response and revised manuscript adequately addressed your concerns. If you have any remaining questions or if there are areas that still require attention, we would greatly appreciate the opportunity to discuss them further.
> >
> > Thank you for your time and consideration.
> >
> > Best regards,
> >
> > Authors

---

> ### Author Response · Authors · 2023-11-18
> **Response to Reviewer eSEd (3/3)**
>
> **Insights of Power Law Distribution**
>
>
> > *"Q2. It’s not clear how the power law with respect to data usage is useful. This power-law may simply represent the underlying distribution of people interested in deep learning topics/domains (E.g. text (NLP) versus mortgage tabular data (Finance))"*
>
>
> The emphasis on precise log-log scale line fitting goes beyond a simple long-tail distribution, offering nuanced insights into dataset usage. We further studied across various tasks and found that **the power law also persists within specific task domain**, despite variations in dataset counts across domains (e.g., 2128 in NLP, 217 in computer vision, 153 in audio, 137 in multimodal, 79 in tabular, with others lacking metadata). This highlights the robustness of the power-law distribution, which can reflect research trends within task domains. Understanding this distribution is crucial for comprehending how datasets are utilized in the machine learning community, aiding resource allocation, and guiding platform providers like Hugging Face in prioritizing high-demand dataset support.
>
>
> We have added a figure to show the power law within different tasks (Figure S5) in Appendix F.
>
>
> **Missing Key Component in Dataset Documentation**
>
>
> > *"Q3. There’s a key component in the dataset documentation that is missing e.g. how the data is processed – perhaps this is covered in data usability?"*
>
>
> There is a section called "Dataset Creation" in the community-endorsed dataset cards, and as the template suggests, it is divided into subsections like:
>
>
> - **(1) Curation Rationale:** This subsection delves into the underlying motivation for creating the dataset, offering insight into its intended purpose and scope.
> - **(2) Source Data:**
>  - **(2.1) Initial Data Collection and Normalization:** Details the methodology employed in gathering and standardizing the initial data.
>  - **(2.2) Who are the source language producers:** Identifies the original creators of the data, whether individuals or systems, providing context about the data's origins.
> - **(3) Annotations:**
>  - **(3.1) Annotation Process:** Describes the procedures and tools used for data annotation.
>  - **(3.2) Who are the annotators:** States whether the annotations were produced by humans or machine-generated.
>
>
> A specific example is the [ncbi_disease](https://huggingface.co/datasets/ncbi_disease#dataset-creation), comprising disease name and concept annotations from 793 fully annotated PubMed abstracts. This dataset exemplifies thorough documentation of its data collection and annotation processes, as demonstrated in the following examples:
>
>
> > **Annotation process**
> >
> > Each PubMed abstract was manually annotated by two annotators with disease mentions and their corresponding concepts in Medical Subject Headings (MeSH®) or Online Mendelian Inheritance in Man (OMIM®). Manual curation was performed using PubTator, which allowed the use of pre-annotations as a pre-step to manual annotations. Fourteen annotators were randomly paired and differing annotations were discussed for reaching a consensus in two annotation phases. Finally, all results were checked against annotations of the rest of the corpus to assure corpus-wide consistency.
>
>
> Despite the critical role of the "Dataset Creation" section in ensuring dataset reproducibility and usability, our analysis reveals a gap in documentation thoroughness. Even in the healthcare domain, where the data collection process is important, only around 23% provide comprehensive details on dataset creation. Our analysis of the varied emphasis across different sections of dataset documentation (See Section 4 and Fig 3 in our paper) also reveals that the "Dataset Creation" section is the second-least detailed section, accounting for merely about 10% of the text in the dataset cards. Given the importance of this dataset creation section in improving dataset usability and reproducibility, we recommend that dataset creators provide more comprehensive documentation of the dataset creation process in our paper.
>
>
> **References**
>
>
> [1] Sixt, Leon, et al. "Do Users Benefit From Interpretable Vision? A User Study, Baseline, And Dataset." International Conference on Learning Representations. 2021.
>
>
> [2] Geiping, Jonas, et al. "How Much Data Are Augmentations Worth? An Investigation into Scaling Laws, Invariance, and Implicit Regularization." The Eleventh International Conference on Learning Representations. 2022.
>
>
> [3] Koch, Bernard, et al. "Reduced, Reused and Recycled: The Life of a Dataset in Machine Learning Research." Thirty-fifth Conference on Neural Information Processing Systems Datasets and Benchmarks Track. 2021.

---

### Official Review · Reviewer_gNxe · 2023-11-11

**Soundness:** 3 good
**Presentation:** 4 excellent
**Contribution:** 2 fair
**Rating:** 8
**Confidence:** 5

**Summary:**

The paper performs an analysis of the quality of dataset cards, a form of data documentation, for 150 datasets on Hugging Face.

**Strengths:**

The motivation and context of the work are clearly explained. Dataset work is an important and undervalued aspect of machine learning research and this paper tackles an important topic given the prolifieration of ML datasets and platforms. The analysis and rationale are mostly presented clearly, such that it could easily be reproduced for another platform. The key insights are clearly highlighted, and they are interesting; the discussion of results is often insightful. The figures are beautiful and the captions well-done and self-contained (this is surprisingly rare, well done!). The discussion, limitations, and significance are well written. The metadata gathered (as far as I can tell) are formatted in a way that will enable future research in this direction.

**Weaknesses:**

Dataset curation, a very relevant domain, is not covered at all in related works. In general the related work lacks earlier works and perspective, e.g. from library science, which could have informed the analysis in the paper in some interesting ways.
The human eval is not given much space compared to the other aspects and I think it could have provided a lot more insights.

**Questions:**

are the values given in the human eval averaged over the 5 annotators (each saw every dataset card?). If so, it would be interesting to see some inter-rater (dis)agreement, and some qualitative demonstration of e.g. which cards were high-agreement sort of gold standard, high agreement bad, and low agreement.

What guidance were the raters given? What level of expertise did they have?

The correlation with content comprehensiveness in the human eval is not super high (~40%), and the rest are all quite low. It would be interesting to compare this with a qualitative description from the annotators of what they found made a better or worse card -- are there factors not detailed here they think it would be more informative to consider?

---

> ### Author Response · Authors · 2023-11-18
> **Response to Reviewer gNxe (1/2)**
>
> We thank reviewer gNxe for their insightful comments and suggestions. We address many of reviewer gNxe's comments in our general response above, and we offer more specific details on certain comments below.
>
> **Related Works**
>
> > *"Dataset curation, a very relevant domain, is not covered at all in related works. In general the related work lacks earlier works and perspective, e.g. from library science, which could have informed the analysis in the paper in some interesting ways."*
>
> We thank reviewer gNxe's comments on improving the related works section. We have added the data curation literature and the library science domain research to the related works section.
>
> **1. Dataset Curation:**
> *"Dataset has long been seen as a significant constraint in the realm of machine learning research [1, 2]. The process of creating datasets remains arduous and time-intensive, primarily due to the costs of curation and annotation [3]. Moreover, the quality of data assumes a pivotal role in shaping the outcomes of machine learning research [4]. Consequently, a profound understanding of datasets is indispensable in the context of machine learning research, and this understanding is most effectively conveyed through comprehensive dataset documentation."*
>
> **2. Library Science:**
> *"In the library domain, numerous works have proposed methods to tackle the absence of universally accepted guidelines for publishing library-linked data. These efforts are aimed at enhancing data quality, promoting interoperability, and facilitating the discoverability of data resources[5-7]"*
>
>
> **Inter-rater Agreement**
>
> > *Q1.are the values given in the human eval averaged over the 5 annotators (each saw every dataset card?). If so, it would be interesting to see some inter-rater (dis)agreement, and some qualitative demonstration of e.g. which cards were high-agreement sort of gold standard, high agreement bad, and low agreement.*
>
> Given a large number of dataset cards, we randomly assigned different sets of dataset cards to annotators to have a larger coverage of dataset cards annotated. To verify the agreement among the annotators, we randomly sampled 30 dataset cards and let the annotators annotate them. We then calculated the inter-annotator agreement using the Intraclass Correlation Coefficient (ICC) and got a value of **0.76**, which is considered a good agreement[8]. Combined with the annotators' solid background in AI research, we consider their annotations to be reliable.
>
>
> **Guidance of Human Evaluation Annotation**
>
>
> > *Q2. What guidance were the raters given?*
>
>
> We illustrated the dataset card evaluation criteria in *Section C.2 Human-Annotated Dataset Card Evaluation Methodology and Criteria* in the Appendix. Annotators were instructed to evaluate the dataset cards on seven aspects, each rated on a scale from 0 to 5. These aspects, derived from previous dataset documentation research and the Hugging Face community's dataset card template, were each accompanied by a detailed description. Furthermore, every score level was clearly defined with illustrative examples.
>
> ---
>
> - ***Structural Organization:** How well is the documentation structured with headings, sections, or subsections?*
> - ***Content Comprehensiveness:** How comprehensive is the information provided in the documentation?*
> - ***Dataset Description:** How effectively does the documentation describe the dataset?*
> - ***Dataset Structure:** How well does the documentation explain the underlying data structure of the dataset?*
> - ***Dataset Preprocessing:** How well does the documentation describe any preprocessing steps applied to the data?*
> - ***Usage Guidance:** How well does the documentation offer guidance on using the dataset?*
> - ***Additional Information:** How well does the documentation provide extra details such as citations and references?*
>
>
> *Each aspect received a score on a scale from 0 to 5, with the following score metrics:*
>
>
> | *Score* | *Description*                               |
> | -------- | -------------------------------------------- |
> | *5*     | *Exceptionally comprehensive and effective* |
> | *4*     | *Very good and thorough*                    |
> | *3*     | *Moderately satisfactory*                   |
> | *2*     | *Insufficient*                              |
> | *1*     | *Poor and inadequate*                       |
> | *0*     | *Absent*                                    |
>
>
> ---

---

> ### Author Response · Authors · 2023-11-18
> **Response to Reviewer gNxe (2/2)**
>
> **Human Annotator Background**
>
> > *Q2. What level of expertise did they have?*
>
> In terms of the background of the annotators, the human annotators were carefully chosen, taking into consideration their expertise and background in the field of AI. Specifically, we selected PhD students with a robust research background in NLP, Computer Vision, Human-AI, Machine Learning, and Data Science. Each annotator possesses extensive experience in utilizing datasets for their own research, thereby ensuring their familiarity with dataset documentation. The annotators' diverse and well-established foundation in AI positions them to comprehend the dataset documentation thoroughly, facilitating the reliable provision of annotations. We have added the selection criteria of human annotators to *Section C.2 Human-Annotated Dataset Card Evaluation Methodology and Criteria* in the Appendix.
>
>
> **Human Evaluation Insights**
>
> > *Q3. The correlation with content comprehensiveness in the human eval is not super high (~40%), and the rest are all quite low. It would be interesting to compare this with a qualitative description from the annotators of what they found made a better or worse card -- are there factors not detailed here they think it would be more informative to consider?*
>
>
> **(1) Correlation Between Dimensions and Overall Quality**
>
> The correlation coefficient between the human evaluation of various dimensions and the overall quality is derived from a linear regression model, which **includes all the evaluated dimensions as variables**. Specifically, each coefficient represents the expected change in Overall Quality in response to a one-unit increase in a corresponding independent dimension, while keeping all other dimensions constant. The 0.39 correlation coefficient between the human evaluation of content comprehensiveness and overall quality, suggests that content comprehensiveness plays a significant role in influencing the overall quality of dataset cards, relative to other dimensions.
> Further emphasizing this point, the Spearman correlation coefficient between the human evaluation of each dimension and overall quality reveals a particularly strong correlation (**0.92**) for content comprehensiveness. This high correlation reinforces the argument that content comprehensiveness is a crucial factor in determining the overall quality of dataset cards.
>
>
> **(2) Qualitative Description**
>
>
> In gathering qualitative feedback, we developed evaluation factors based on the dataset card standards endorsed by the Hugging Face community. Annotators concurred that the current dimensions of evaluation are comprehensive and cover the necessary aspects. However, they noted that not all dimensions hold equal importance and that their relevance may vary according to the dataset's domain. For instance, the "Considerations for Using the Data" dimension is more critical for medical datasets compared to others.
>
>
> **References**
>
>
> [1] Halevy, Alon, Peter Norvig, and Fernando Pereira. "The unreasonable effectiveness of data." IEEE intelligent systems 24.2 (2009): 8-12.
>
>
> [2] Sun, Chen, et al. "Revisiting unreasonable effectiveness of data in deep learning era." Proceedings of the IEEE international conference on computer vision. 2017.
>
>
> [3] IBM. "Overcome obstacles to get to ai at scale." https://www.ibm.com/downloads/cas/VBMPEQLN, 2020.
>
>
> [4] Liang, Weixin, et al. "Advances, challenges and opportunities in creating data for trustworthy AI." Nature Machine Intelligence 4.8 (2022): 669-677.
>
>
> [5] Villazón-Terrazas, Boris, et al. "Methodological guidelines for publishing government linked data." Linking government data (2011): 27-49.
>
>
> [6] Hidalgo-Delgado, Yusniel, et al. "Methodological guidelines for publishing library data as linked data." 2017 International Conference on Information Systems and Computer Science (INCISCOS). IEEE, 2017.
>
>
> [7] Abida, Rabeb, Emna Hachicha Belghith, and Anthony Cleve. "An end-to-end framework for integrating and publishing linked open government data." 2020 IEEE 29th International Conference on Enabling Technologies: Infrastructure for Collaborative Enterprises (WETICE). IEEE, 2020.
>
>
> [8] Koo, Terry K., and Mae Y. Li. "A guideline of selecting and reporting intraclass correlation coefficients for reliability research." Journal of chiropractic medicine 15.2 (2016): 155-163.

---

> ### Author Response · Authors · 2023-11-21
> **We would like to hear back from Reviewer gNxe**
>
> Dear Reviewer gNxe,
>
> We would like to follow up to see if our response and revised manuscript address your concerns or if you have any further questions. Should there remain any points that require further attention, we would be grateful for the chance to engage in further discussion.
>
> Thank you for your time and consideration.
>
> Best regards,
>
> Authors

---

### Author Response · Authors · 2023-11-18
**Response to all reviewers (1/2)**

We thank the reviewers for their thoughtful and constructive evaluation of our manuscript. We were greatly encouraged by the reviewers' recognition of the novelty and significance of the discussed topic (Reviewers gNxe, kkaF), as well as their acknowledgment of the comprehensiveness of our conducted analysis (Reviewers gNxe, kkaF, ALk3) and the insightful findings and their good presentation (Reviewers gNxe, eSEd, ALk3). Additionally, we appreciate the affirmation of the practical utility of the practice we propose (Reviewers kkaF, ALk3) and the good cross-evaluation by human annotators (Reviewer ALk3). In response to the invaluable feedback received, we present a general response to address points raised by multiple reviewers. Furthermore, individualized responses are provided below to effectively address each reviewer's specific concerns. We have carefully revised and enhanced the manuscript in accordance with the received feedback. The changes are highlighted in the revised manuscript.

- Power Law in Dataset Usage (Section 2): We have included new figures (Figure S5) in Appendix F to illustrate the power law distribution within each task domain to highlight the robustness of the power law distribution across different task domains.

- Humain Evaluation Annotation (Section 6): We detailed the selection process and verification of human annotators in Appendix C.2, emphasizing their extensive AI expertise and the high agreement among them (ICC=0.76). This addresses the concerns raised by reviewers gNxe and ALk3.

- Related Works (Section 7): We have significantly expanded our literature review section to provide additional information on data curation and research from the library science domain.

- Section Extraction Method (Appendix C): We validated the reliability of the exact keyword matching method in Appendix C.1, showcasing a notably high coverage rate of 84% in mapping the extracted titles to those suggested by the Hugging Face community.

- Section Fill-out Rate by Downloads (Appendix F): We have included a figure (Figure S6) in Appendix F to illustrate the fill-out rate of each section by dataset popularity.

- Diverse Documentation Practices among Creators (Appendix F): We included a table detailing the characteristics of dataset documentation by curators from varied backgrounds to address reviewer kkaF's query.

- Applicability to Other Platforms (Appendix G): We have added an analysis of dataset cards on GitHub utilizing the methodology outlined in our paper, demonstrating the transferability of our approach across different platforms in response to Reviewer ALk3's concern.

In response to the reviewer eSEd's concern about **the work contribution and relevance to ICLR**, we would like to emphasize the following points:


- **Data-Centric AI Research is a core part of ML:** The evolving focus of the Machine Learning community on Data-centric AI is evident in the recent trends at major conferences, including ICLR. Significantly, ICLR has introduced **"datasets and benchmarks"** as an explicit topic this year, reflecting its growing interest in this domain. Previous ICLR conferences have accepted studies analyzing dataset practices, such as [1] and [2]. Furthermore, NeurIPS, another leading ML conference, started a separate Datasets and Benchmarks Track in 2021. Notably, [3] conducted a comprehensive dataset analysis and was recognized with a best paper award. While there is an abundance of research on analyzing and creating datasets, our work uniquely addresses the under-explored area of dataset documentation practices, underscoring its relevance and importance to the ML community and ICLR.
- **Importance of Dataset Documentation in ML research:**
Dataset documentation is a crucial yet often overlooked aspect of machine learning research, pivotal for ensuring the reliability, reproducibility, and transparency of ML datasets. Our study makes a meaningful contribution in this area by presenting a thorough analysis of current dataset documentation practices. The insights and findings from our investigation are not just academically interesting but carry practical implications as well. We have identified key gaps in existing dataset documentation and proposed specific improvements, offering actionable advice for both practitioners and the broader community. A notable suggestion from our work is the introduction of a 'usage section' in dataset card templates for the Hugging Face community. This recommendation aligns with ICLR's focus on impactful research that advances the field of machine learning, emphasizing the significance and suitability of our study for the conference.

---

> ### Author Response · Authors · 2023-11-18
> **Response to all reviewers (2/2)**
>
> Regarding reviewer ALk3's query about the **applicability of our findings to other platforms**, we would like to highlight that our choice of the Hugging Face platform as a case study was strategic. This decision was based on two key factors: (1) Hugging Face's status as one of the most popular ML platforms, and (2) its unique position as one of the few open-source platforms providing an official dataset template. We believe in the transferability of our study's method, especially considering that README files are a prevalent documentation format broadly employed across various platforms. To facilitate further research, we have made available the code for data retrieval and analysis, as well as the dataset, at https://anonymous.4open.science/r/HuggingFace-Dataset-Card-Analysis. We also included a case study applying our proposed method to the dataset cards on GitHub, illustrating the transferability of our analysis to different platforms. Our research provides a robust foundation for future investigations into documentation practices on other platforms, underscoring the potential for wide applicability of our findings.
>
> In summary, our study uncovers the current community norms and practices in dataset documentation and demonstrates the importance of comprehensive dataset documentation in promoting transparency, accessibility, and reproducibility in the AI community. We hope to offer a foundation step in the large-scale empirical analysis of dataset documentation practices and contribute to the responsible and ethical use of AI while highlighting the importance of ongoing efforts to improve dataset documentation practices.
>
> We would again like to thank all reviewers for their time and feedback, and we hope that our changes adequately address all concerns.
>
> **Reference**
>
> [1] Sixt, Leon, et al. "Do Users Benefit From Interpretable Vision? A User Study, Baseline, And Dataset." International Conference on Learning Representations. 2021.
>
> [2] Geiping, Jonas, et al. "How Much Data Are Augmentations Worth? An Investigation into Scaling Laws, Invariance, and Implicit Regularization." The Eleventh International Conference on Learning Representations. 2022.
>
> [3] Koch, Bernard, et al. "Reduced, Reused and Recycled: The Life of a Dataset in Machine Learning Research." Thirty-fifth Conference on Neural Information Processing Systems Datasets and Benchmarks Track. 2021.

---

### Meta-Review · Area_Chair_4L3V · 2023-12-07

**Metareview:**

A) This paper does a thorough meta-analysis of datasets on huggingface, and gives useful insights on the state of dataset documentation as whole in 2023.
B) Reviewers all agreed the paper is well written and tackles an important problem, does it well, provides clear guidance on what dataset authors can do to create better dataset cards and a general template for "successful" dataset cards
C) The paper is a meta-analysis of the field so does not fix any issue with dataset cards but points out how to fix future dataset cards. Additionally many of the paper's findings are intuitive and do not shed any new insight into the state of datasets.

Despite this the reviewers seem to agree the paper is a good fit for ICLR so I vote for acceptance.

**Justification For Why Not Higher Score:**

While paper is good, reviewers do seem split on how actionable the findings are, and how dataset authors can better adapt practices from this work.

**Justification For Why Not Lower Score:**

The paper is very well written and tackles an important problem and all reviewers seem to agree about this.

---

### Decision · Program_Chairs · 2024-01-16

Accept (poster)